# Ubx-Collier signaling cascade maintains blood progenitors in the posterior lobes of the *Drosophila* larval lymph gland

**Aditya Kanwal**[1], **Pranav Vijay Joshi**[1], **Sudip Mandal**[2], **Lolitika Mandal**[1]*

1 Developmental Genetics Laboratory, Department of Biological Sciences, Indian Institute of Science Education and Research (IISER) Mohali, SAS Nagar, Punjab, India, 2 Molecular Cell and Developmental Biology Laboratory, Department of Biological Sciences, Indian Institute of Science Education and Research (IISER) Mohali, SAS Nagar, Punjab, India

* lolitika@iisermohali.ac.in

**Data Availability Statement:** All relevant data are within the manuscript and its Supporting Information files.

## Abstract

*Drosophila* larval hematopoiesis occurs in a specialized multi-lobed organ called the lymph gland. Extensive characterization of the organ has provided mechanistic insights into events related to developmental hematopoiesis. Spanning from the thoracic to the abdominal segment of the larvae, this organ comprises a pair of primary, secondary, and tertiary lobes. Much of our understanding arises from the studies on the primary lobe, while the secondary and tertiary lobes have remained mostly unexplored. Previous studies have inferred that these lobes are composed of progenitors that differentiate during pupation; however, the mechanistic basis of this extended progenitor state remains unclear. This study shows that posterior lobe progenitors are maintained by a local signaling center defined by Ubx and Collier in the tertiary lobe. This Ubx zone in the tertiary lobe shares several markers with the niche of the primary lobe. Ubx domain regulates the homeostasis of the posterior lobe progenitors in normal development and an immune-challenged scenario. Our study establishes the lymph gland as a model to tease out how the progenitors interface with the dual niches within an organ during development and disorders.

## Author summary

Stem and progenitor cells require a niche for their maintenance. Invertebrate models have played a pivotal role in deciphering the signals/crosstalks between stem cell compartments and their niches. The multilobed larval hematopoietic organ or the lymph gland of *Drosophila* has been extensively used to gain insights into the mechanistic basis of these interactions.

The lymph gland consists of cellularly diverse primary lobes, followed by pairs of secondary and tertiary lobes, collectively called posterior lobes. These less explored posterior lobes that houses blood progenitors is the current focus of the study. Based on molecular markers, we propose that the posterior lobes comprise blood cells in a pre-progenitor-like state and an anterior domain in the tertiary lobe that expresses Hox gene Ultrabithorax (Ubx) and high EBF factor Collier/Knot. Employing loss of function, genetic knockdown

**Funding:** A.K. received Senior Research Fellowship from the Department of Biotechnology (DBT), Government of India [DBT/JRF/15/AL/270] (www.dbtjrf.gov.in). P.V.J. recieved DST-INSPIRE Scholarship for Higher Education (www.online-inspire.gov.in). S.M. is funded by IISER Mohali Institutional support (www.iisermohali.ac.in). L.M. received DBT/Wellcome Trust India Alliance (www.indiaalliance.org) Senior Fellowship [IA/S/17/1/503100]. The funders had no role in study design, data collection and analysis, decision to publish, or preparation of the manuscript.

**Competing interests:** The authors have declared that no competing interests exist.

and overexpression analysis, we demonstrate that the Ubx domain maintains the posterior lobe progenitors' state and fate.

Hox gene Antennapedia is known to specify the niche of the primary lobe. Our results elucidate a similar Hox-dependent signaling center essential for maintaining the posterior lobes' blood progenitors. This study raises the possibility of using the lymph gland to explore how multiple niches interact with progenitors during development and disorders.

## Introduction

Blood cells in *Drosophila* bear several similarities to vertebrates in development and function [1,2]. The hematopoietic events, giving rise to mature blood cells, occur in two spatially and temporally distinct *Drosophila* developmental phases. The first wave, known as primitive hematopoiesis, occurs in the head mesoderm of early embryos [1] whereas, the second wave, known as definitive hematopoiesis, initiates in the cardiogenic mesoderm region of late embryos [3], which is akin to the AGM (aorta-gonad-mesonephros) of the vertebrates.

*Drosophila* larval lymph gland, an outcome of AGM hematopoiesis [3], is a primary hemato-poietic site in larval stages. It comprises proliferating blood progenitors developing inside a bi-laterally symmetrical multi-lobed organ consisting of primary, secondary, tertiary, and rarely quaternary lobes [4–6]. The anterior/primary lobe consists of three defined zones, a niche (posterior signaling center; [7–9], medullary zone (that includes pre-progenitors and intermediate progenitors [10–14]), and the cortical zone housing differentiated hemocytes [10]. The cells in the cortical zone consist of plasmatocytes, crystal cells, and lamellocytes; the latter formed only upon immune challenge; [1,15]. The primary lobe has been a subject of tremendous investigations revealing several intricate crosstalks between the three distinct zones [2,7,8,10,16]. In contrast, the posterior lobes (comprising secondary and tertiary lobes) of this multi-lobed structure remain comparatively less explored. Few reports which delved into the nature of posterior lobes commented on their homogenous nature and the variance in terms of their size from one genotype to another [10,17–20] or highlighted the heterochrony observed in the developing lymph gland lobes [21]. A recent report also emphasizes that the posterior lobes consists of blood progenitors [22].

While differentiation is observed in the primary lobe progenitors as early as the second instar [10], posterior lobe cells differentiate only in the pre-pupal stages [23]. However, these posterior lobes respond to immune challenges by initiating excessive proliferation (hyperplasia), precocious differentiation, and lamellocyte induction [17,21] during late larval stages.

Although several extrinsic and intrinsic signals are critical for maintaining the primary lobe progenitors [2], the mechanistic basis of this prolonged progenitor hood evident in the posterior lobes is yet to be answered.

Here in this study, we have characterized these posterior lobes and found a unique pattern of zonation signifying distinct cell types within these lobes. Along with a bulk of progenitors, a domain in the anterior half of the tertiary lobe has characteristic high Collier (Col) and Ultra-bithorax (Ubx) expression. Our temporal analysis reveals that Ubx expresses in the entirety of the posterior lobes during early larval instars and restricts with Col to the anterior half of the tertiary lobe with development. Sustenance of Ubx expression depends on Col, which is also intrinsically required to maintain posterior lobes' progenitors. This unique Ubx expressing domain shares several markers with the niche of the primary lobe and is crucial for the maintenance of posterior lobe progenitors both during development and immune challenges. Thus, this study provides a comprehensive characterization and functional analyses of an uncharacterized Hox domain within the posterior lobes that serves as a signaling center for the posterior lobes' progenitor pool.

## Results

### The cells in the posterior lobes are heterogeneous

The lymph gland posterior lobes are believed to be a reservoir of progenitors kept aside for post-larval requirements [2,17,24]. Previous studies have concluded that the posterior lobes consist of homogeneous cell type (Fig 1A). The idea of homogeneity stems from the fact that upon using markers like DE-Cadherin or Shotgun (*shg*; [10]), a uniform expression is evident in the posterior lobes when compared to primary lobes (Fig 1B, 1B′ and 1B‴). Moreover, terribly reduced optic lobe (Trol; [25]), a heparin sulfate proteoglycan seems to wrap up the entire posterior lobes (Fig 1B, 1B″ and 1B‴).

In order to have a better insight, we initiated a marker-based analysis using the primary lobe progenitors as reference cell type. Studies on the primary lobe have shown that the progenitor pool within the developing lymph gland is heterogeneous. While Angiotensin Converting Enzyme (Ance; [26,27]) is a pan progenitor marker (Fig 1C and 1C′), Pvf2 (a ligand for Pvr: platelet-derived growth factor; [28,29]) expression defines the pre-progenitors within the Ance expressing pool [27]. Interestingly, posterior lobes are high in both of these proteins, indicating that most of the cells are in a progenitor-like state [Fig 1D–1D‴]. However, in all lymph glands analyzed, the anterior half of the tertiary lobe remains negative for Ance (denoted by an asterisk in Fig 1C′ and 1D′). Coincidentally, the expression of bonafide progenitor markers, Domeless [10] and TepIV [30] are not uniform in these lobes (Fig 1E–1F″); instead, many cells in the anterior half of the tertiary lobe were reproducibly negative for the bonafide progenitor markers (Fig 1E″ and 1F″, marked by an asterisk).

Based on these data, we inferred that the anterior domain of the tertiary lobe does not harbour any pre-progenitors or progenitors. However, this does not rule out the presence of intermediate progenitors in this domain that might be primed to differentiate upon immune challenge. To address this aspect, we first analyzed the posterior lobes' hemocytes for differentiation markers at 96hr AEH (post 48hr infection [31] with *Leptopilina boulardi* [32]). Interestingly, in all cases, the anterior domain of the tertiary lobe (marked by an asterisk) lacked differentiation marker Peroxidasin (Pxn, [27,33,34]), ruling out the possibility of IP cells (Fig 1G–1H′).

This observation led us to infer that the posterior lobes of the lymph gland are majorly composed of progenitors but also have an uncharacterized domain in the anterior part of the tertiary lobe (white cells in Fig 1I). The cells within this domain neither express pan progenitor markers nor give rise to differentiated cells upon immune challenge.

To delve further into the nature of this domain, we focused our attention on extracellular matrix proteins. As ECM components such as Trol [25] and Viking [10] have been known to express heterogeneously in the lymph gland's primary lobe, we assayed for their detailed expression pattern in posterior lobes using GFP fusion constructs. While the expression inside the lobes is minimal, intriguingly, both these proteins divide the secondary and tertiary lobes into two domains (S1A–S1H′ Fig, domains marked by asterisks). This compartmentalization indicates a possibility of mutually exclusive and distinct functional units within the posterior lobes. To address this, we employed a previously uncharacterized Gal4; 103908-Gal4 [35] active in the posterior domain of both the secondary and tertiary lobes (S1I–S1J′ Fig). Activation of the lineage-tracing cassette [36] by the 103908-Gal4 from the very beginning fails to mark the anterior domain (S1K–S1L′ Fig). Thus, it is clear that the two domains are mutually exclusive. The above result, coupled with our expression studies, suggests that the anterior domain cells of the tertiary lobes are different in their state and fate compared to the rest of the reserve progenitors.

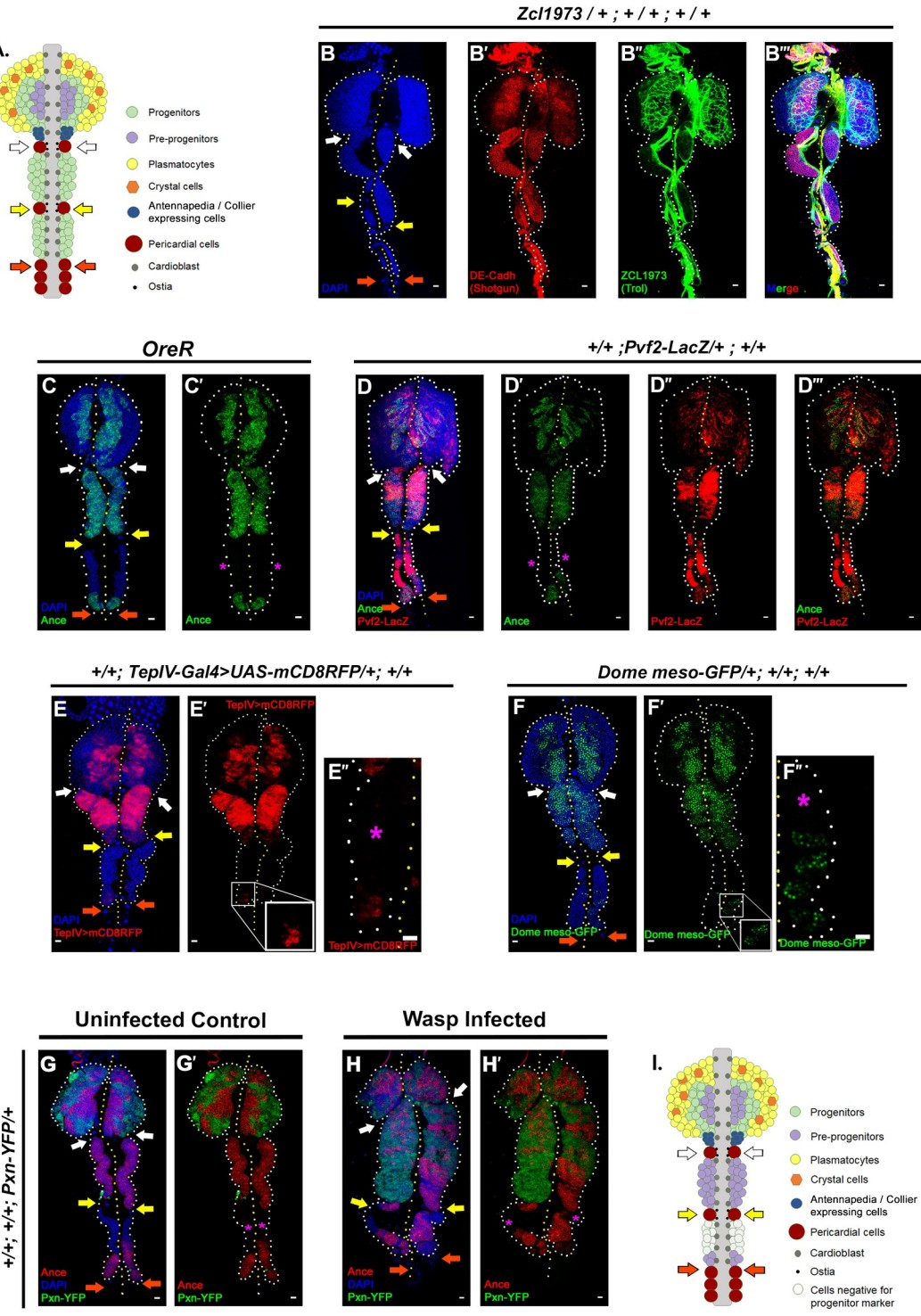

**Fig 1. Posterior lobes house heterogenous cell populations.** (A.) Model depicting current understanding of third instar lymph gland. (B-B′″.) Shotgun (DE-Cad) is homogenously expressed in the posterior lobes (B′), while Trol wraps up the entire posterior lobes (B″). (C-C′.) Except for the anterior half of the tertiary lobe, the cells of the posterior lobes express Ance, a pan progenitor marker. (D-D′″.) Co-expression of Ance and Pvf2 defines the pre-progenitors of posterior lobes. (E-F″.) Bona-fide marker of the lymph gland progenitors, TepIV, and Domeless express uniformly but at different levels in distinct regions of secondary lobes. Cells in the anterior half of the tertiary lobe lack TepIV and Dome expression (E″-F″, magenta asterisk), but some cells in the posterior half do (zoomed-in inset). (G-H′.) The anterior half (marked with a magenta asterisk) does not express progenitor markers such as Ance or differentiated markers such as Pxn, even upon

immune challenge via *Leptopilina boulardi* (H-H') in comparison to control (G-G'). (I.) Model summarizing our findings. In all panels, lymph glands are dissected from third instar larvae (96hr after egg hatching, AEH) unless mentioned otherwise. Arrows are pointed towards the intercalated pericardial cells within lymph gland lobes. The white arrow marks the pericardial cell between primary and secondary lobes; yellow between secondary and tertiary, and orange marks the last pericardial cell at the end of the LG organ after the tertiary lobe. White dotted lines mark the lymph gland boundary, while the yellow dotted line marks the aorta, which is between the bi-lobed lymph gland. The genotype of the larvae and labeling on the tissue are described in the panels. Scale bars: 20 μm. See S1 Fig.

These observations prompted us to undertake a detailed characterization of the anterior domain of the tertiary lobe.

## Ubx and Collier expression changes dynamically in the posterior lobes during development

The posterior lobes are known to express the gene *collier/knot*, which encodes the *Drosophila* orthologue of mammalian early B-cell factor (EBF) [22,37–39], essential for B-cell lymphopoiesis in mice [40,41]. To visualize Col expression, we first employed two of the available Gal4: pCol85-Gal4 [7] and Knot[13A11]-Gal4 (GMR13A11/ Kn[13A11]-Gal4: [42,43]) and co-stained them with Col Antibody [44] at 24, 48, 72, 96 AEH and in pre-pupa. Our expression analysis reveals that Kn[13A11]-Gal4 domain co-localizes with high Col expression throughout development (S2A–S2I'' Fig). Similar co-localization can be observed with pCol85-Gal4 post 48hrs AEH (S2B–S2J'' Fig). It is important to note that the posterior lobes express the Hox gene Ubx [22]. To further characterize the posterior lobes, we initiated a detailed expression analysis of Ubx throughout development.

Co-labelling of Ubx with Knot (visualized by Kn[13A11]-GAL4>UAS-mCD8RFP) expression was performed at 24, 48, 72, 96, 110 hr AEH and prepupa (Figs 2A–2E'''' and S3A–S3D'''). Both Ubx and Col/Knot express in the newly formed posterior lobes and the intercalating pericardial cells at 24hr AEH (Fig 2A–2A''''). However, by 48hr AEH, Ubx is restricted to the tertiary lobes compared to Kn[13A11]-Gal4, which is more widespread in its expression (Fig 2B–2B''''). Beyond 48hr AEH, the expression of Ubx gets further limited to the anterior half of tertiary lobes, which can be much appreciated by 72hr AEH (Fig 2C–2C''''). At this time point, the Knot expression level is diminished in the secondary lobes while highly persisting in the tertiary lobes.

At 96 hr AEH, a complete withdrawal of Kn[13A11]-Gal4 expression is evident in the secondary lobes, while in the tertiary lobe Ubx and Knot majorly co-localize (Fig 2D–2D''''). Around 110hr AEH Ubx further regresses, leaving an area low in Knot expression (asterisk in S3A–S3B''' Fig). By 120hr AEH, this dynamicity of Ubx and Knot is most evident (Fig 2E–2E''''), revealing distinct regions from which Ubx has regressed (asterisks in Fig 2E'''–2E''''). This restriction of Ubx and Kn[13A11]-Gal4 expression with time is suggestive of the existence of specific domains within the tertiary lobe. Intrigued to know the state of these cells, we labelled them with progenitor (Ance, S3A–S3B''' Fig) as well as differentiated cell (Pxn, S3C–S3D''' Fig) markers. The low Knot cells that lack Ubx expression are positive for Ance (magenta asterisk, S3A–S3B''' Fig), thereby demonstrating that the Ubx domain in the tertiary lobe lacks progenitors at this developmental stage. On the other hand, the Knot domain (inclusive of Ubx expressing cells) does not express differentiated cell marker Pxn (S3C–S3D''' Fig, magenta asterisk).

This dynamicity of Col/Knot expression seems to be essential to carve out the domain of progenitors analogous to its similar function in the primary lobe [39].

To assess whether Col is also required for progenitor maintenance of the posterior lobes akin to primary lobes, we employed *col¹ (kn^{col-1} / CyO; P{col5-cDNA.C} / TM6B)*, a Collier

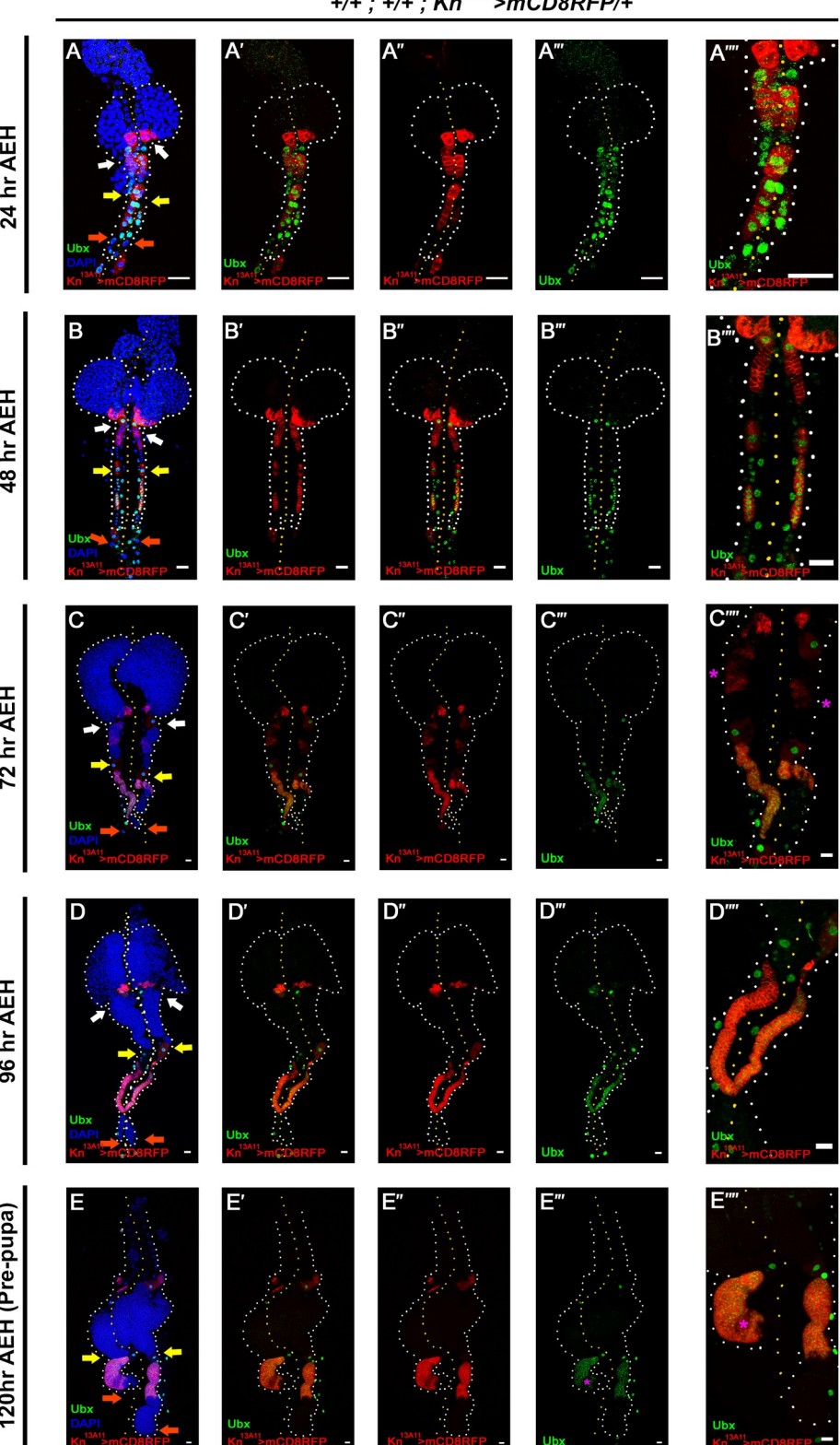

**Fig 2. Ultrabithorax (Ubx) and Collier expression changes dynamically in the posterior lobes.** (A-A"".) At 24hr AEH, the entire posterior lobes comprise a single stripe of Ubx and Col expressing blood cells. (B-B"".) By 48 hr AEH, the Ubx expression in the lymph gland is restricted in the tertiary lobe, while Col expression persists in secondary

lobes. (C-C"".) At 72 hr AEH, Ubx forms a distinct domain in the tertiary lobe's anterior part while Col expression is residual in the secondary lobe (magenta asterisk in C""). (D-D"".) At 96hr AEH, in the posterior lobes Col expression is only in the tertiary lobe and colocalizes with Ubx. (E-E"".) By 120 hr, Ubx further restricts to carve out a region low Col expression (magenta asterisk). Larval staging is mentioned in the corresponding panels. Arrows are pointed towards the intercalated pericardial cells within lymph gland lobes. The white arrow marks the pericardial cell between primary and secondary lobes; yellow between secondary and tertiary, and orange marks the last pericardial cell at the end of the LG organ after the tertiary lobe. White dotted lines mark the lymph gland boundary, while the yellow dotted line marks the aorta, which is between the bi-lobed lymph gland. The genotype of the larvae and labeling on the tissue is described in the panels. Scale bars: 20 μm. See S2 and S3 Figs.

null allele [45–47]. Posterior lobes from the *col*[1] homozygous undergo precocious differentiation (S3F–S3G′ Fig), which otherwise does not differentiate by 96hr AEH (S3E and S3E′ Fig).

Previous studies on the lymph gland's primary lobe have shown that Hox gene Antennapedia (Antp) expression fails to sustain in the absence of Col/Knot [8]. Whether Ubx expression is also dependent on Col was next investigated. Although Ubx can be seen in the pericardial cells and cardioblast of *col*[1] larvae (marked by an asterisk), a complete lack of Ubx is seen in the lymph gland (S3I–S3J′ Fig), in comparison to wild type control (S3H and S3H′ Fig).

These data collectively implicate that the dynamic expression of Ubx and Col throughout development is essential to define domains within the posterior lobes. Additionally, it is evident that Ubx expression depends on Col function for its sustenance. This relationship recapitulates an identical scenario wherein Hox gene Antp also requires Col for its maintenance in the hematopoietic niche of the primary lobe [8].

## Ubx expressing domain in late third instar does not house progenitors

Massive differentiation is observed in the posterior lobes in response to wasp parasitism in *Drosophila* larvae by 96 hr AEH [17,39]. However, no substantial response was observed at 72 hr AEH of LG (24 hr post-infection), as reported previously [48], (Fig 3A and 3B″).

By 96hr AEH (48 hr post-infection), a downregulation in progenitor marker Ance was observed, and there was an onset of differentiation marker Pxn (Fig 3C–3D″). Primary lobes were observed to dissociate in most of the cases. Interestingly, the anterior half of the tertiary lobe, which has characteristic high Col and Ubx expression, does not differentiate. This zone neither expresses progenitor markers such as Ance nor differentiates (lacks Pxn) upon parasitic challenge.

To further explore the nature of Ubx positive cells of the tertiary lobe, we screened for several markers. Ubx domain that does not express either progenitors or differentiated cell markers is housed in a separate compartment visualized by Trol expression (S4A–S4A‴ Fig). Similar compartmentalization of the primary lobe niche by Trol [25] prompted us to speculate if the Ubx domain expresses markers similar to the Antp expressing PSC/niche of the primary lobe. Intriguingly, several PSC markers express in the Ubx positive zone, more notably Hedgehog [8] known to be released from niche to the neighbouring progenitors in primary lobe, expresses in comparatively lower levels in the Ubx zone (Fig 3E–3G), visualized by *hhF4fGFP*, a transcriptional reporter of *hh* [49]. A similar level of Dad, a readout of *dpp* signaling [50], is also detectable in the Ubx domain (Fig 3H–3J), visualized by dad-nRFP [51]. Additionally, a few cells of the Ubx domain express Serrate [52] (Ser, magenta asterisk, S4B and S4B′ Fig), Lamin C [13] (LamC, magenta asterisk, S4C and S4C′ Fig) and Fringe connection (Frc, magenta asterisk, S4D and S4D′ Fig), in the late third instar.

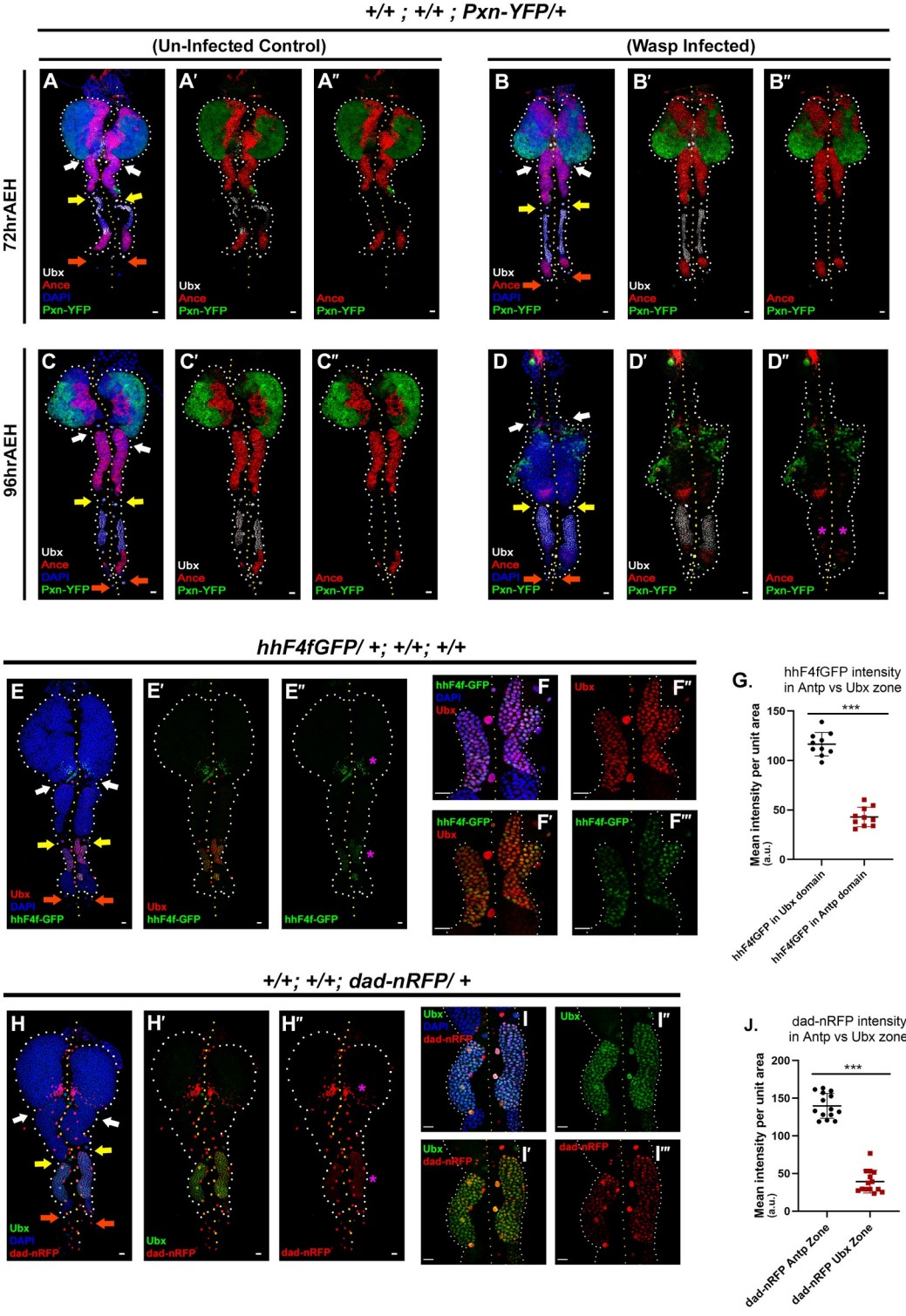

**Fig 3. Posterior Lobe harbours a unique cell population expressing bonafide markers of primary lobe PSC.** (A-B".) No significant response against parasitic challenge (B-B") is observed in posterior lobes at 72 hr AEH (24 hr post-infection) in comparison to control (A-A"). (C-D".) Differentiation and hyperplasia are observed in the posterior lobes upon wasp parasitism by 96hr AEH (48hr post-infection) (D-D") compared to control (C-C"). Ubx zone neither expresses Pxn nor Ance upon wasp infection (magenta asterisk, D"). (E-F"'.) *hhF4fGFP* expression in the primary lobe PSC and the anterior part of the tertiary lobe (magenta asterisk, E-E"). Higher magnification revealing the co-localization of Hh and Ubx (F-F"'). (G.) Intensity analysis of *hhF4fGFP* reveals the differences in Hedgehog expression between Antp and Ubx zone (n = 10, P-

value = 2.040E-11). (H-I'''.) Dad expression in the primary lobe niche and the anterior half of tertiary lobes colocalizes with Ubx (magenta asterisk, H-H') and zoomed-in images (I-I'''). (J.) Quantitative analysis reflecting the difference in the intensity of Dad between Antp and Ubx zone (n = 14, P-value = 2.238E-15). In all panels, lymph glands are dissected from third instar larvae (96hr after egg hatching, AEH) unless mentioned otherwise. Arrows are pointed towards the intercalated pericardial cells within lymph gland lobes. The white arrow marks the pericardial cell between primary and secondary lobes; yellow between secondary and tertiary, and orange marks the last pericardial cell at the end of the LG organ after the tertiary lobe. White dotted lines mark the lymph gland boundary, while the yellow dotted line marks the aorta, which is between the bi-lobed lymph gland. The genotype of the larvae and labeling on the tissue is described in the panels. Scale bars: 20 μm. Two-tailed unpaired Student's t-test was performed for calculating statistical significance. Data are mean±s.d. *P<0.05, **P<0.001 and ***P<0.0001. See S4 Fig.

Put together, our results suggest that the Ubx domain in the late third instar lymph gland defines a unique population of hemocytes that shares several markers with the hematopoietic niche (PSC) of the primary lobe (S4E Fig).

## The progenitor pool of the posterior lobes depends on the Ubx domain for their maintenance

The restriction of Ubx in the anterior domain of the tertiary lobe prompted us to address the significance of this domain to the rest of the posterior lobes. More importantly, the wrapping of Ubx domain by ECM similar to the primary lobe PSC [11] raises the possibility of communication between ECM-separated posterior lobe cells.

*Ubx¹*, a null allele of Ubx [53], is embryonic lethal, and the level of Ubx in the posterior lobe of the heterozygotes is comparable to similarly aged control. Therefore, to create a hypomorphic condition specific to our tissue of interest, we combined *the GAL4-UAS* system with this amorph following the timeline shown in Fig 4A. An RNAi mediated down-regulation of *Ubx* by pCol-Gal4 was performed in the background of *Ubx¹*. The majority of the tertiary lobes from the lymph gland of this genotype exhibited a significant drop in Ubx levels compared to control (Fig 4B–4G' and quantitated in 4K).

Due to the decline in the level of Ubx in the anterior domain of the tertiary lobe, a precocious differentiation is noticeable mostly in secondary lobes and in several examples of tertiary lobes (Pxn, compare Fig 4B and 4B' with 4D and 4D' and 4F and 4F', and quantitated in 4H–4J). It is to be noted that this genetic manipulation did not result in ectopic differentiation in the primary lobes (Fig 4H) and the Ubx expression in neighbouring cells such as the cardioblasts and pericardial cells is unaltered. Intriguingly, in cases where partial Ubx expression persists in hypomorph, lower differentiation is observed (Fig 4D', magenta asterisk).

In the prepupal stages (4-5hr APF), the posterior lobes differentiate and dissociate [23]. Interestingly in genotypes where *Ubx* is overexpressed by pCol-Gal4, even after 5hr APF, intact secondary and tertiary lobes can be seen (Fig 4L–4M').

Since pCol-Gal4 also expresses in the primary lobe niche, we wanted to test out the role of Ubx with a posterior lobe specific driver. We narrowed down to Ubx-Gal4^M1 [54] in search of one such driver. Continuous activation of the lineage-tracing cassette [36] with Ubx-Gal4 ^M1 revealed that only the posterior lobes are lineage traced. At the same time, the active expression is limited to the anterior domain of the tertiary lobe at 96 hr [S5A–S5A'' Fig]. Interestingly, this restricted expression of Ubx–Gal4^M1 coincides with Ubx antibody 96 hr AEH (S5B–S5C''' Fig). Downregulation of Ubx employing Ubx-Gal4^M1 following the temperature regime (Fig 5A) evoked differentiation in the otherwise undifferentiated posterior lobes (Pxn, Fig 5B–5F). Interestingly, analysis of prepupal samples revealed that the upregulation of *Ubx* by Ubx-Gal4^M1 delayed the rupture of the lymph gland (Fig 5G and 5H). Using two independent drivers to downregulate the levels of Ubx clearly demonstrates that the Ubx domain has the potential to maintain the state of the progenitors.

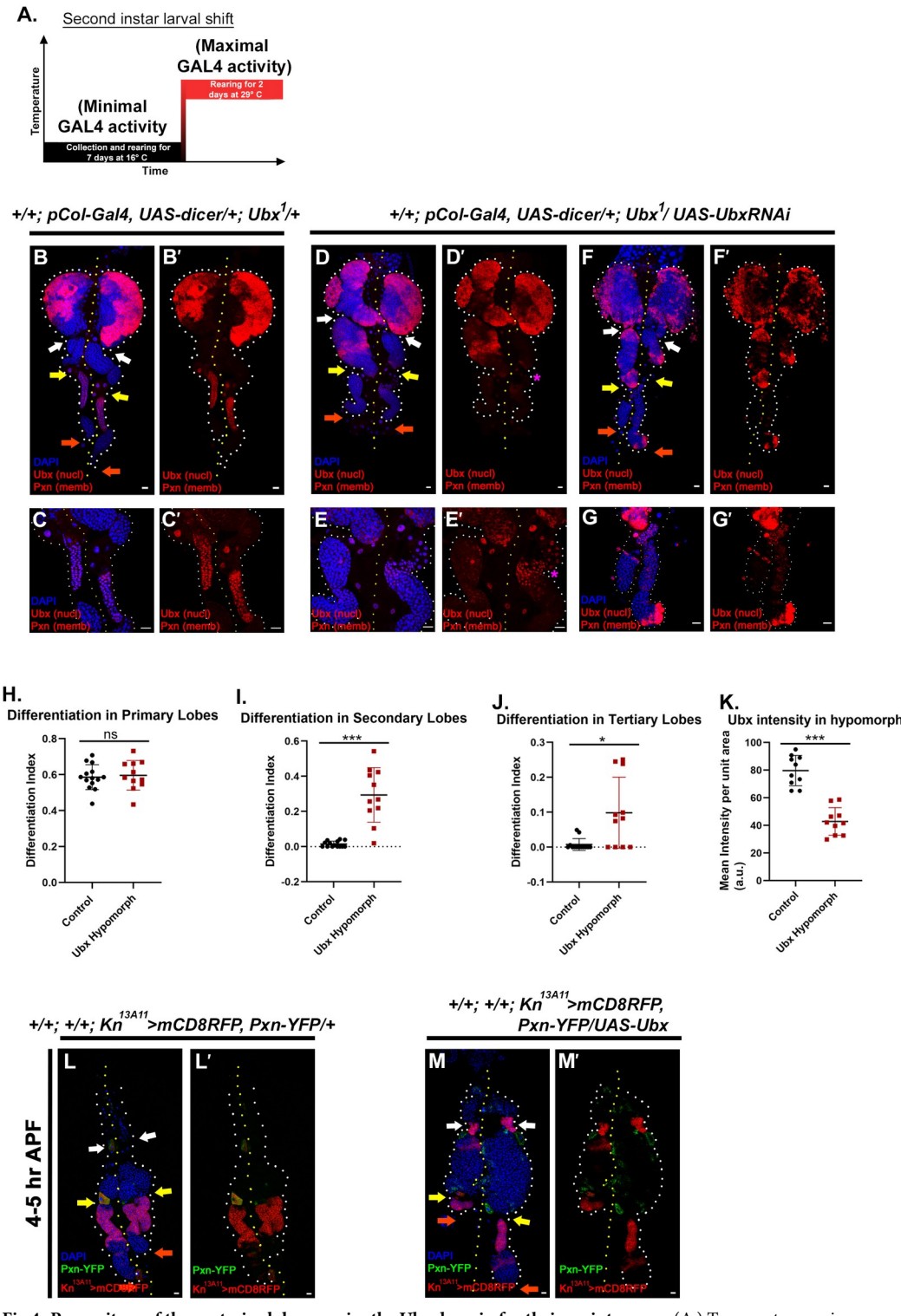

**Fig 4. Progenitors of the posterior lobes require the Ubx domain for their maintenance.** (A.) Temperature regime adopted for the experiments performed in this panel. (B-G') Compromised Ubx levels in Ubx hypomorphs results in ectopic differentiation (Pxn) observed in the posterior lobes (D-G') compared to controls (B-C'). Zoomed in images reveal the drop in Ubx levels (C-C', E-E', and G-G'). (H.) Quantification of the differentiation index of primary lobe (B-G') {n = 14 (control), n = 11 (hypomorph), P-value = 0.7479}. (I.) Quantification of the differentiation index of the secondary lobe (B-G') {n = 14 (control), n = 11 (hypomorph), P-value = 1.32E-04). (J.) Quantification of the differentiation index of tertiary lobe progenitor (B-G') (n = 14 (control), n = 11 (hypomorph), P-value = 1.44E-02). (K.) Quantification of Ubx

levels in the anterior domain of tertiary lobe of hypomorph (B-G') {n = 10, P-value: 3.431E-07}. (L-M'.) Over-expression of Ubx delays the secondary lobe's differentiation and dispersion (M-M') as late as 4hr after pupal formation (APF) compared to control (L-L'). In all panels, lymph glands are dissected from third instar larvae (96hr after egg hatching, AEH) unless mentioned otherwise. Arrows are pointed towards the intercalated pericardial cells within lymph gland lobes. The white arrow marks the pericardial cell between primary and secondary lobes; yellow between secondary and tertiary, and orange marks the last pericardial cell at the end of the LG organ after the tertiary lobe. White dotted lines mark the lymph gland boundary, while the yellow dotted line marks the aorta, which is between the bi-lobed lymph gland. The genotype of the larvae and labeling on the tissue is described in the panels. Scale bars: 20 μm. Two-tailed unpaired Student's t-test was performed for calculating statistical significance. Data are mean±s.d. *P<0.05, **P<0.001 and ***P<0.0001.

Moreover, the downregulation of *Ubx* from the anterior domain of the tertiary lobe (Fig 5I–5K) revealed that with the loss of Ubx, Col declines (Fig 5L–5N), indicating that Ubx function is genetically upstream of Col.

This observation is analogous to the relationship of Antp and Col observed in the PSC of the primary lobe [8]. Our next step was to explore whether the Ubx domain in the posterior lobes emanates maintenance signals like the PSC of the primary. In order to address our hypothesis, the first approach adopted was downregulation of *sar1 (secretion–associated Ras-related1*, [55]) in order to block the communication of this domain with the neighbouring progenitors. Sar1 regulates the formation of coat protein complex II (COPII), which is involved in transporting the newly synthesized proteins from the endoplasmic reticulum to the Golgi [56–58]. Downregulating *sar1*, thus, leads to the intracellular accumulation of proteins that are otherwise meant to be released extracellularly [59,60]. Downregulation of *sar1 function by* Kn[13A11]-Gal4 following the scheme (S6A Fig) resulted in a drastic increase in differentiation and overall size of the lymph gland (S6B–S6C' Fig, and quantitated in S6E–S6G Fig).

Increased differentiation of the posterior lobes is also evident when Sar1 is downregulated by Ubx-Gal4[M1] following the scheme (S6H Fig). As the Ubx-Gal4[M1] is active only in the posterior lobes (S6I–S6J' Fig and quantitated in S6L–S6N Fig), differentiation in the anterior lobe is comparable to the control. The above result suggests that secretory molecules emanating from the Ubx domain essentially regulates the posterior lobe progenitors' maintenance.

We wondered whether Hh, which expresses in the same cells, serves as the maintenance signal. Employing both drivers, it was clear that the loss of Hh from the Ubx domain leads to ectopic differentiation (Compare S6B and S6B' with S6D and S6D', S6E–S6G Fig; and S6I and S6I' with S6K and S6K', S6L–S6N Fig).

These results collectively suggest that the Ubx domain regulates the fate and state of the progenitor cells of the posterior lobes. Given the requirement of Ubx for posterior lobe homeostasis, we designate this domain as the posterior lobe signaling center (PLSC).

## Ubx domain/PLSC responds to wasp parasitism and induces lamellocyte formation

To gain further insight into the Ubx domain/ PLSC's functionality, we assayed its response to immune challenges. Upon infecting the second instar larvae with parasitic wasps (*Leptopilina boulardi*), ectopic differentiation, hyperplasia, and lamellocyte induction were observed in the posterior lobe progenitors (Fig 6A–6B' and 6D–6J). In sync with earlier studies [17,48], the primary lobe was either highly differentiated or completely dispersed in many cases. On the other hand, secondary lobe progenitors also exhibited massive amounts of proliferation and ectopic differentiation along with lamellocyte induction (Figs 6A–6B' and 6D–6J and S7A–S7B', S7D–S7E' and S7G–S7M). There was almost no differentiation in the few available progenitors of the tertiary lobe. Moreover, the level of Ubx from PLSC was substantially

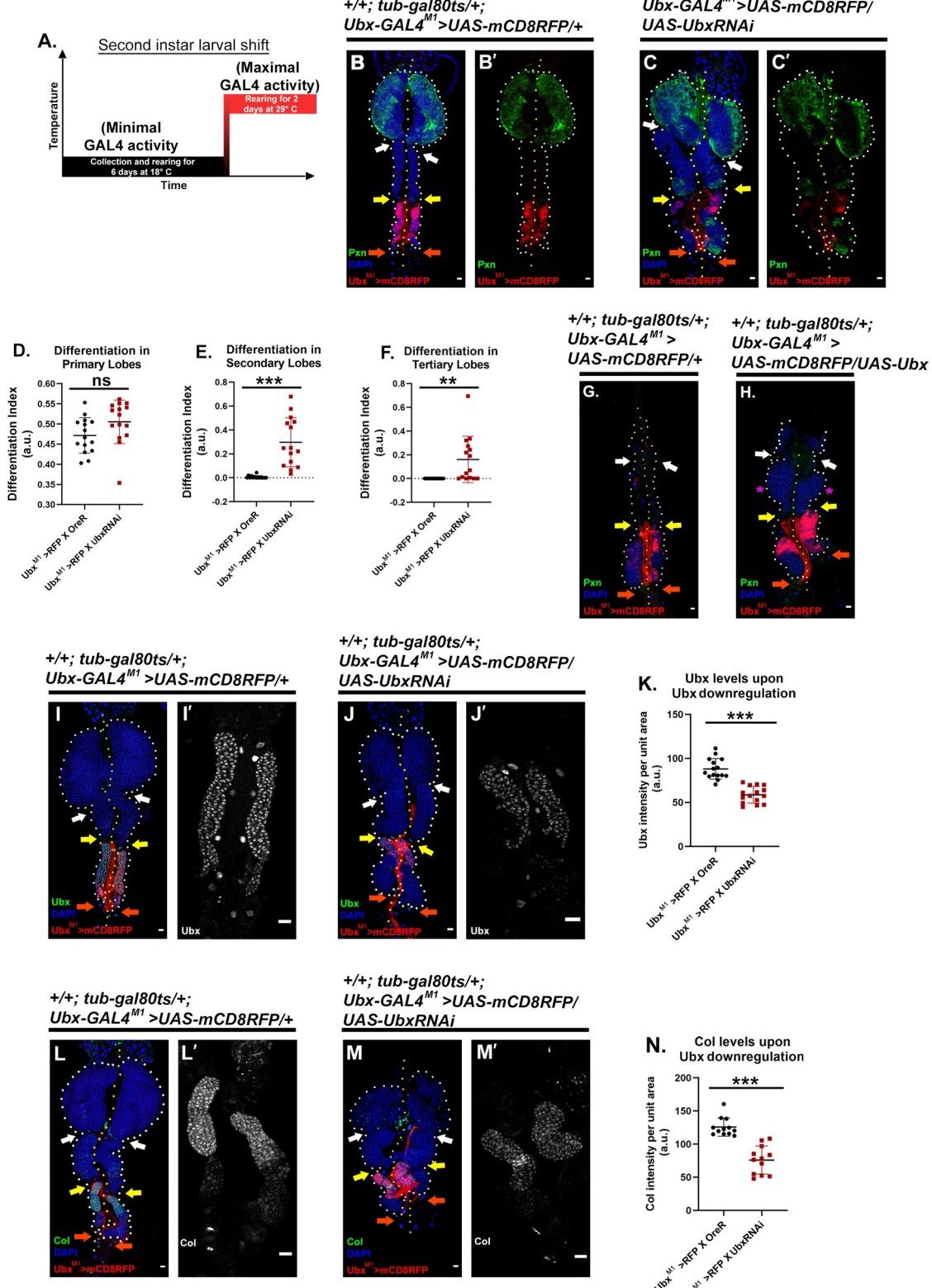

**Fig 5. Posterior lobe specific driver endorses the requirement of Ubx domain for the maintenance of posterior lobe progenitor.**
(A.) Temperature regime followed for experiments in this figure. (B-C') Downregulation of *Ubx* via posterior lobe specific driver,

Ubx-Gal4$^{M1}$ results in ectopic differentiation (Pxn) observed in the posterior lobes (C-C') compared to controls (B-B'). (D.) Quantification of the differentiation index of primary lobe (B-C') {n = 15, P-value = 0.0697}. (E.) Quantification of the differentiation index of the secondary lobe (B-G') {n = 15, P-value = 7.87E-05}. (F.) Quantification of the differentiation index of tertiary lobe progenitor (B-G') {n = 15, P-value = 6.64E-03}. (G-H.) Over-expression of *Ubx* via posterior lobe specific driver, Ubx-Gal4$^{M1}$ delays the secondary lobe's differentiation and dispersion (magenta asterisk) as late as 4hr after pupal formation (APF). (I-J'.) Reduction in Ubx levels is evident upon *Ubx* downregulation from posterior lobe specific driver. (K.) Quantification of Ubx levels in the anterior domain of tertiary lobe of hypomorph (I-J'.) {n = 15, P-value: 4.259E-08}. (L-M'.) Reduction in the Col levels is evident upon *Ubx* downregulation from posterior lobe specific driver. (N.) Quantification of Col levels in the anterior domain of tertiary lobe of hypomorph (L-M') {n = 12, P-value: 1.671E-06}. In all panels, lymph glands are dissected from third instar larvae (96hr after egg hatching, AEH) unless mentioned otherwise. Arrows are pointed towards the intercalated pericardial cells within lymph gland lobes. The white arrow marks the pericardial cell between primary and secondary lobes; yellow between secondary and tertiary, and orange marks the last pericardial cell at the end of the LG organ after the tertiary lobe. White dotted lines mark the lymph gland boundary, while the yellow dotted line marks the aorta, which is between the bi-lobed lymph gland. The genotype of the larvae and labeling on the tissue is described in the panels. Scale bars: 20 μm. Two-tailed unpaired Student's t-test was performed for calculating statistical significance. Data are mean±s.d. *P<0.05, **P<0.001 and ***P<0.0001. See S5 and S6 Figs.

downregulated upon wasp infection (S8A–S8C Fig), further endorsing the importance of Ubx during wasp infection.

On upregulation of *Ubx* from PLSC by Kn$^{13A11}$-Gal4, no ectopic differentiation, or lamellocyte induction in the secondary lobes, was observed post wasp infection. However, in this genotype, plenty of lamellocyte wrapped the primary lobe (Fig 6C, 6C' and 6D–6J). Overexpression of *Ubx* with PLSC specific Ubx-Gal4$^{M1}$ gave similar results (S7C, S7C', S7F, S7F', and S7G–S7M Fig). These observations implicate that the PLSC/Ubx domain can modulate posterior lobe progenitors during an immune challenge.

To have a mechanistic insight into the above observations, we assayed the status of Hh upon overexpression of *Ubx* by Ubx-Gal4$^{M1}$ /Kn$^{13A11}$-Gal4, which resulted in the upregulation of Hh levels assayed by *hhF4fGFP* (S9A–S9H Fig).

Blocking the communication between progenitors and the PLSC either by downregulation of *sar1*, *hh* or *Ubx* resulted in enhanced immune response upon Wasp challenge (S10A–S10J Fig), which in many cases led to smaller posterior lobes due to precocious dispersal of hemocytes. Additionally, we also demonstrated that perturbation of Hh signaling from the PLSC elicits differentiation in the progenitors (S6K and S6K' Fig). Together, these findings implicate that Hh is one of the maintenance signals required for the posterior lobe progenitor homeostasis.

Studies have illustrated that the Antp positive PSC/niche in the primary lobe plays a crucial role in generating an immune response. In addition to EGFR/Spitz signaling, a simultaneous upregulation of the Toll/NFκB pathway by the niche evokes lamellocyte formation in response to wasp infection [61,62]. Tempted to see if the PLSC acts similarly, we assayed the status of D4-LacZ (a reporter for Dif activation). Interestingly, the PLSC, which, unlike primary lobe PSC, does not express Dif even at basal levels in uninfected controls (S11A, S11A' and S11F Fig), initiates a high level of Dif expression upon wasp infection (S11B–S11C' and S11F Fig). Since the overexpression of *Ubx* prevented lamellocyte induction in the posterior lobes upon infection, we hypothesized that this genotype might have prevented the activation of D4 in the posterior lobes. Indeed, overexpression of *Ubx* via pCol-Gal4 resulted in a drop in Dif levels in the PLSC (S11D, S11D' and S11F Fig). Interestingly, this genetic manipulation did not interfere with the Dif activation in the primary lobe's PSC (S11E Fig).

During development, ROS levels in the primary lobe progenitors are crucial for their differentiation [34,63]; however, it is barely detectable in the niche [61]. We assayed ROS levels in the posterior lobes by DHE and ROS-inducible-GST-promoter-GFP (*gstD-GFP*;[64]). In both cases, a low level of ROS is evident in the Ubx domain compared to the neighbouring progenitors (S11G–S11G″, S11J and S11J' Fig). In contrast, a higher ROS and increased *gstD-GFP*

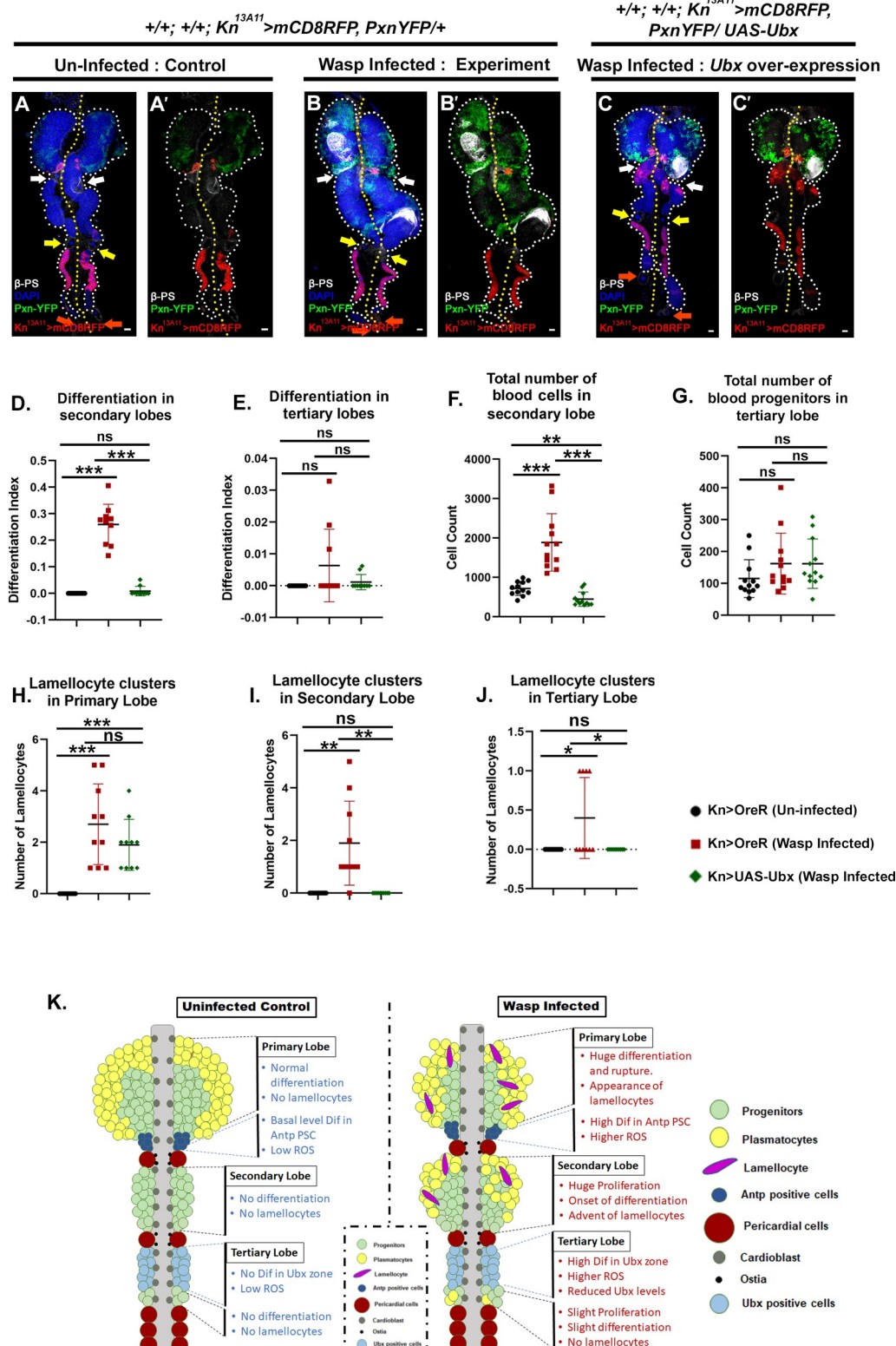

**Fig 6. Ubx domain enables posterior lobe progenitors to mount an immune response.** (A-C'.) Ectopic differentiation and lamellocyte induction were observed upon wasp infection in primary and secondary lobes (B-B') compared to uninfected control (A-A'). Overexpression of *Ubx* from the tertiary lobe suppresses the immune response from the posterior lobes (C-C'). (D.) Quantification of the differentiation index of secondary lobe (A-C'.) {Uninfected vs Infected:

n = 10, P-value = 1.889E-06}, {Uninfected vs *Ubx* overexpression: n = 10, P-value = 0.171} and {Infected vs *Ubx* overexpression: n = 10, P-value = 1.376E-06}. (E.) Quantification of the differentiation index of tertiary lobe (A-C'.) {Uninfected vs Infected: n = 10, P-value = 0.114}, {Uninfected vs *Ubx* overexpression: n = 10, P-value = 0.169} and {Infected vs *Ubx* overexpression: n = 10, P-value = 0.190}. (F.) Quantification of total hemocyte pool in secondary lobe (A-C') {Uninfected vs Infected: n = 12, P-value = 1.51E-04}, {Uninfected vs *Ubx* overexpression: n = 12, P-value = 1.19E-03} and {Infected vs *Ubx* overexpression: n = 12, P-value = 2.120E-05}. (G.) Quantification of the progenitor pool in tertiary lobe (A-C'.) {Uninfected vs Infected: n = 12, P-value = 0.164}. {Uninfected vs *Ubx* overexpression: n = 12, P-value = 0.110} and {Infected vs *Ubx* overexpression: n = 12, P-value = 0.998}. (H.) Quantification of lamellocytes in primary lobe (A-C') {Uninfected vs Infected: n = 10, P-value = 4.06E-04}, {Uninfected vs *Ubx* overexpression n = 10, P-value = 1.92E-04} and {Infected vs *Ubx* overexpression: n = 10, P-value = 0.192}. (I.) Quantification of lamellocytes in secondary lobe (A-C') {Uninfected vs Infected: n = 10, P-value = 4.43E-03} and {Infected vs *Ubx* overexpression: n = 10, P-value = 4.43E-03}. (J.) Quantification of lamellocytes in tertiary lobe (A-C'.) {Uninfected vs Infected: n = 10, P-value = 3.67E-02}, and {Infected vs *Ubx* overexpression: n = 10, P-value = 3.67E-02}. (K) Lymph gland model summarizing the response of LG against immune challenge. In all panels, lymph glands are dissected from third instar larvae (96hr after egg hatching, AEH) unless mentioned otherwise. Arrows are pointed towards the intercalated pericardial cells within lymph gland lobes. The white arrow marks the pericardial cell between primary and secondary lobes; yellow between secondary and tertiary, and orange marks the last pericardial cell at the end of the LG organ after the tertiary lobe. White dotted lines mark the lymph gland boundary, while the yellow dotted line marks the aorta, which is between the bi-lobed lymph gland. The genotype of the larvae and labeling on the tissue is described in the panels. Scale bars: 20 μm. Two-tailed unpaired Student's t-test was performed for calculating statistical significance. Data are mean±s.d. *P<0.05, **P<0.001 and ***P<0.0001. See S7, S8, S9, S10 and S11 Figs.

levels were observed in the Ubx domain/PLSC upon wasp infection, in a manner identical to that of Antp expressing PSC/niche (S11H–S11H″, S11I, S11K, S11K′ and S11L Fig).

The above experiments implicate that the PLSC/Ubx domain plays an integral role in enabling the posterior lobe progenitors to mount an immune response (summarized in Fig 6K) against the wasp challenge.

## Col expressing Ubx negative blood progenitors of the posterior lobes contributes toward the Adult hematopoietic hub

An earlier study has shown that the progenitors from the posterior lobe contribute towards the adult hematopoietic hub [24]. We decided to perform a similar experiment with Ubx-Gal4$^{M1}$, which upon lineage tracing marks the entire posterior lobe (S5A–S5A″ Fig). Indeed, upon continuous activation of the lineage tracing construct with Ubx-Gal4$^{M1}$ till 96 hrs AEH (Fig 7A), it is evident that hematopoietic hub harbours a substantial number of Ubx lineage traced cells (Fig 7B, 7C and summarized in Fig 7D). Based on the above observation we can infer that the lineage- traced cells in the hub in adult arise from the posterior lobe progenitors, which have subsequently differentiated in the adult (P1$^{+ve}$, Fig 7B and 7C).

Co-expression studies with Ubx and Col establish that the widespread expression of Col and its subsequent restriction to the Ubx domain carves out a discrete progenitor pool, which can be visualized as Ance$^{+}$, Ubx$^{-}$ and Col$^{-}$ (S3B–S3B‴ Fig). With the identification of the Col expression in the Ubx domain and the insight that Col expression undergoes a time-dependent restriction, we revisited the previous observation [24] to dissect the contribution of Col expressing progenitors to adulthood. We activated the same lineage tracing cassette (G-TRACE; [36]) at a different time point, in sync with dynamic Col/Kn expression (i.e. 72hrAEH, 96hr AEH, and 120hr AEH (refer Fig 2C–2E″″)). The hematopoietic hub was then subjected to detailed analysis for the Knot expressing lineage traced cells five days post eclosion.

Upon activation of the lineage-tracing cassette by Kn$^{13A11}$-Gal4 at 72hr AEH (Fig 7E), a substantial number of Knot/Col positive cells were lineage traced in the adult hematopoietic hub (Fig 7F–7F″ and 7K), which is in agreement with the earlier study. However, on activation of the lineage-tracing cassette at a later time point (96hr AEH; Fig 7G), the number of lineage traced cells declines (Fig 7H–7H″ and 7K). Interestingly, post-120 AEH, activation of the

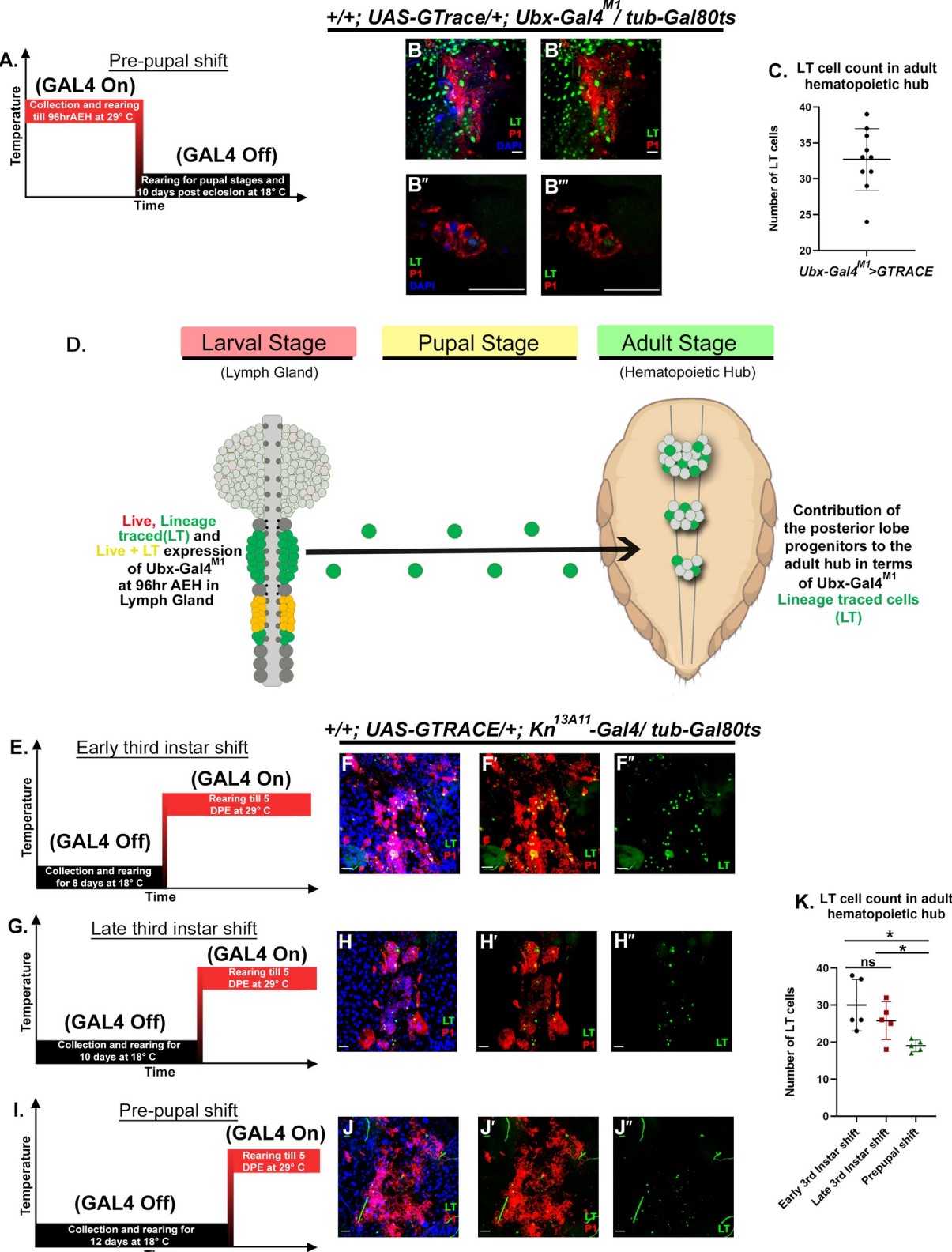

**Fig 7. Posterior Lobes contribute blood cells to the adult hematopoietic hub.** (A) Timeline for the lineage-tracing experiment followed (B-B'''), where the posterior lobe specific driver, Ubx-Gal4[M1] is employed for lineage tracing. The average number of cells getting lineage traced are depicted in graph (C). (D) Infographic representation of the contribution of posterior lobes progenitors to the adult hematopoietic hub. (E)

Timeline for the lineage-tracing experiment shown in (F-F") where an average of 30 lineage traced cells co-localizing with P1 were observed in the adult hub. (G.) Timeline for the lineage-tracing experiment shown in (H-H"). An average of 26 lineage-traced cells co-localizing with P1 were observed in the adult hub. (I) Timeline for the lineage-tracing experiment followed (J-J") With a late shift, an average of 19 lineage traced cells positive for P1 were observed in the adult hub (K.) Quantitative analyses denoting a decrease in the number of lineage traced cells in the adult hub with the change in shifting regime. {Early 3rd instar shift vs Late 3rd instar shift: n = 5, P-value = 0.3116}, {Early 3rd instar shift vs Pre-pupal shift: n = 5, P-value = 0.0224} and {Late 3rd instar shift vs Pre-pupal shift: n = 5, P-value = 0.0384}. Hematopoietic hubs in all panels are dissected from the adult fly five days post eclosion. The genotype of the larvae and labeling on the tissue is described in the panels. Scale bars: 20 μm. Two-tailed unpaired Student's t-test was performed for calculating statistical significance. Data are mean±s.d. *P<0.05, **P<0.001 and ***P<0.0001.

cassette (Fig 7I) lineage traces only a few cells indicating that a restricted number of progenitor cells are Col positive at this time point in the posterior lobes (Fig 7J–7J″ and 7K).

Put together, our results establish that posterior lobe progenitors indeed contribute to the hematopoietic hub in the adult fly. Furthermore, it is intriguing to see that the posterior lobe experiences a wave of Col before it restricts to the PLSC/Ubx domain. Therefore, the dynamic regression of Col from the progenitors explains the decrease in the number of Col lineage-traced cells observed in the hematopoietic hub through our time-dependent lineage tracing studies.

## Discussion

The multi-lobed lymph gland that spans from thoracic to abdominal segment with its stockpile of progenitors at different developmental stages provides us with a unique opportunity to unravel signals dedicated to their maintenance. The extrinsic, intrinsic, and systemic signals critical for maintaining the primary lobe progenitors have been extensively worked out [2,16,59]. In comparison, our understanding of the cells in the posterior lobe is limited. Previous studies have described the cells of the posterior lobes as progenitor populations reserved for post-larval hematopoiesis [17,24]. These lobes grow in size due to the robust cell proliferation during the late larval stages. At this time point, the lymph gland consists of G2 arrested primary lobe progenitors [27] and the proliferating posterior lobe progenitors [10].

Our expression and genetic analysis of the posterior lobe progenitors in the *Drosophila* lymph gland identifies a unique domain/PLSC that majorly caters to the maintenance signal for the reserve population. It is to be noted that Col expression in the primary lobe regresses to the PSC/Antp expressing hematopoietic niche with time, leaving behind a tiny subset of Col positive progenitors in the mature third instar lymph gland [2,39,65,66]. Our findings unravel a similar scenario in the posterior lobes, wherein the widespread expression of Col in the progenitors gradually restricts to the Ubx domain at later stages. Similar to the primary lobe where Col expression sustains Antennapedia expression in the PSC/niche [8], we show that Col in the posterior lobes is essential for Ubx expression in the PLSC. Loss of Col thus affects progenitor maintenance both intrinsically as well as extrinsically through the Hox domains in the lymph gland. In conjunction with the previous study [8], our findings illustrate that the HOX–Collier axis governs these two otherwise distinct larval hemocyte progenitors for their maintenance (Fig 8). With the identification of the maintenance potential of the Ubx domain/PLSC in the tertiary lobe, it can be inferred that the two distinct progenitors of the lymph gland are dependent on two niches that are laid down by the Hox code.

The posterior lobe progenitors that seem to tailgate the primary lobe progenitors during development provide us with an exciting yet unexplored area to unravel signals that create this developmental delay. Our study reveals that the Ubx domain in the mature lymph gland's posterior lobe expresses Hedgehog, which seems essential for progenitor maintenance.

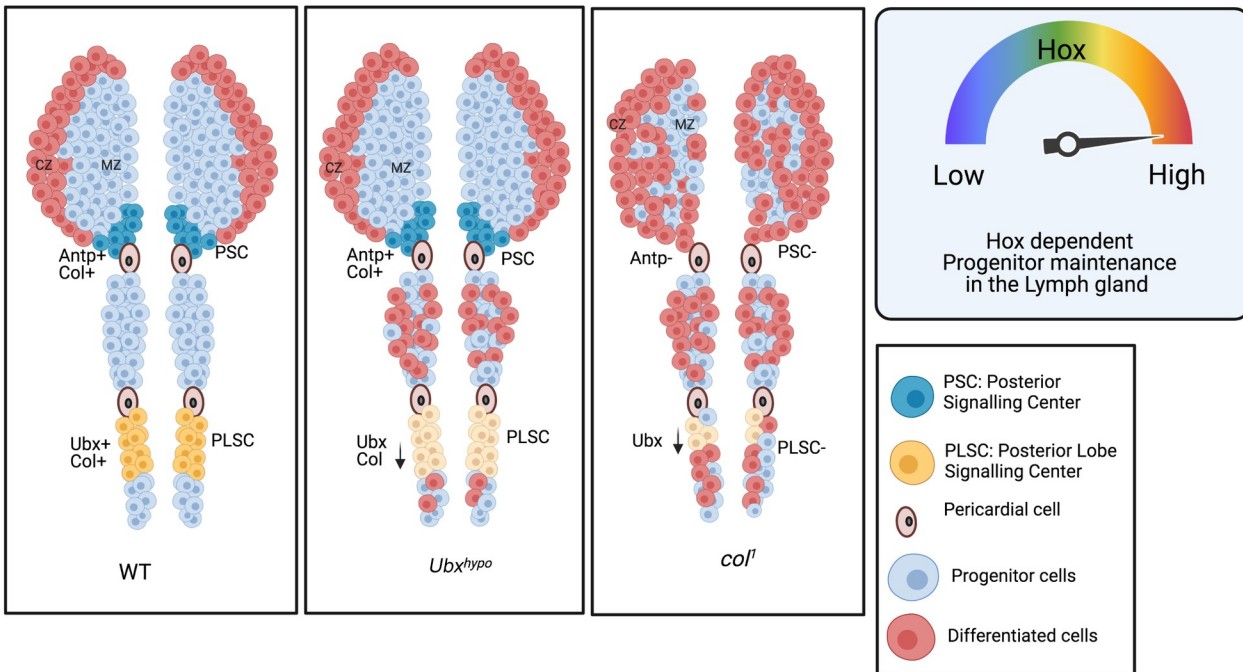

**Fig 8. Collier collaborates with Hox for the maintenance of the lymph gland progenitors.** The PSC of the primary lobe is specified by the Hox gene Antp and maintained by Col. The newly identified signaling center in the posterior lobes PLSC is specified by Hox gene Ubx and also maintained by Col. Downregulation of *Ubx* or *col* in the PLSC pushes the posterior lobe progenitors towards precocious differentiation.

The secondary lobe is the largest of all the posterior lobes and houses abundant progenitors. Interestingly, not before pupation, this progenitor pool differentiates. We believe that this prolonged progenitor hood can be attributed to its position between the two HOX domains: Antp expression PSC to its anterior and the Ubx expression PLSC to its posterior. Sandwiched between two Hh expressing domains the secondary lobe progenitors survives all differentiation signal during larval life.

Therefore, this temporal lag in differentiation is physiologically relevant to the animal. The prolonged progenitor-hood endows the cells the potential to participate in post-larval hematopoiesis. However, during immune challenges, this pool is recruited precociously to mount a response.

## Materials and methods

### Fly stocks

In this study, the following *Drosophila* strains were used: Zcl1973 (L. Cooley), Pvf2-LacZ (M. A. Yoo), Dome meso-GFP, Viking-GFP, UAS-GTRACE, Ser9.5-LacZ (U. Banerjee), *hhF4fGFP* (R.A. Schulz), dad-nRFP (M. Gonzales-Gaitan), pCol85-Gal4, UAS-Dicer; D4 lacZ (A. Sharma), gstD-GFP (D. Bohmann), pCol85-Gal4 (M. Crozatier) and Ubx-Gal4[M1] (E. Sanchez-Herrero, L. S. Shashidhara). The following stocks were procured from Bloomington *Drosophila* Stock Center: OreR, UAS-mCD8RFP, UAS-2XeGFP, Kn[13A11]-Gal4, UAS-tubgal80ts, UAS-UbxRNAi and UAS-sar1RNAi. UAS-hhRNAi was procured from Vienna *Drosophila* Resource Center and following stocks were procured from KYOTO Stock Center: Pxn-YFP, TepIV-Gal4, 103908-Gal4, Frc-Gal4, UAS-Ubx, Col[1] and Ubx[1]. Detailed genotype of the fly lines used for the current work is listed in S1 Table. Kn[13A11]-Gal4 (III) and pCol-Gal4 (II) {detailed genotypes in S1 Table} were observed to have same expression pattern. pCol-Gal4

was used for the convenience of genetics wherever genetic manipulations were not possible due to chromosomal restrictions.

For developmental time series experiments, larvae were synchronized as follows: 2-hour collections were taken after two discard collections of 1 hour each. Newly hatched larvae within a 1 hour interval the next day were collected and transferred onto fresh food plates and aged for specified time periods at 25° C or 29° C depending on the experiment. All stocks were maintained at 25°C on standard media.

### Immunohistochemistry

Lymph gland dissection and immunostaining was performed using previously described protocols [8,10].

The following primary antibodies were used: mouse anti-Ubx [1:10, FP3.38, Developmental Studies Hybridoma Bank (DSHB)], mouse anti-Antp (1:10,8C11, DSHB), mouse anti-LaminC (1:20,LC28.26, DSHB mouse anti-P1 (1:40, gift from I. Ando, Hungarian Academy of Sciences, Szeged, Hungary; [67]), rat anti-E-cadherin (1:50, DCAD2, DHSB), mouse anti-β-PS (1:5, CF.6G11, DSHB), rabbit anti-β-galactosidase (1:100, A11132, Molecular Probes), rabbit anti-Ance (1:500, A. D. Shirras; [68]), mouse anti-Pxn (1:500, gift from J. Fessler) and guinea pig anti-Col (1:1000, gift from S. Thor).

Secondary antibodies used in this study were: mouse-Cy3 (115-165-166), mouse-FITC (315-097-003), rabbit-Cy3 (711-165-152), rat-Cy3 (712-165-153), mouse-647 (115-605-003) and rabbit-647 (711-605-152). All used in 1:500 dilution, procured from Jackson ImmunoResearch Laboratories.

Tissues were mounted in Vectashield (H-1000, Vector Laboratories) then followed by Confocal Microscopy (Leica SP8 and Zeiss LSM780). Blocking was carried out using 5% normal donkey serum (Jackson ImmunoResearch Laboratories, 017-000-121).

### Detection of ROS

Larvae were dissected in Schneider's medium (Gibco, 21720001) and incubated in 0.3 μM DHE (Molecular Probes, D11347) prepared in Schneider's medium, for 8 minute at room temperature in dark. This was followed by two washes in 1X PBS for 5 minute each; a brief fixation was done with 4% PFA for 10 min followed by two 5 minute 1X PBS washes. Tissues were then mounted in Vectashield and imaged in confocal microscope.

### Infecting larvae with Wasp *Leptopilina boulardi*

Wasps used for infection studies were a kind gift from T. Mukherjee and the infections were performed following the protocol mentioned in [31]. To summarize, 6–8 females and 6–8 males were transferred to food vials containing 48hr AEH staged larvae for 4 hours. Wasps were then removed from the vials and infected larvae were reared at 25° C or 29° C depending on the experiment.

### Imaging and statistical analyses

All images were captured as z-sections using Leica SP8 and Zeiss LSM780 confocal microscope. Same settings were used for biological and technical replicates. All the experiments were repeated at least thrice (biological replicates) to ensure reproducibility.

## Cell counting

Cells were counted using the spot function in Imaris software http://www.bitplane.com/download/manuals/QuickStartTutorials5_7_0.pdf) [27]. Data are expressed as mean±s.d. of values from three sets of independent experiments. All statistical analyses were performed using the two-tailed Student's t-test.

## Calculating differentiation index

The diameter of nuclei in each sample was found out using the Slice tool in Imaris. By adding the nuclear diameter into the Spot tool, the total number of cells (based on the number of nuclei) in all stacks of a specified lobe was counted by the program's inbuilt algorithm. The resultant spots were crosschecked manually to co-localize with all the sample nuclei by adjusting the threshold. Filter for the channel that marked differentiated cells was then added in the Spot tool window to count the number of differentiated cells in all the stacks of the same set. This was again crosschecked manually to mark all the differentiating cells by adjusting the threshold. Number cells expressing differentiated cell marker divided by the total number of cells in the specified lobe yields differentiation index.

## Quantification of intensity analysis

Intensity analysis of cytoplasmic signals and stainings in different genotypes were carried out using the protocol mentioned in [69] (https://celldivisionlab.files.wordpress.com/2012/06/measuring-cell-fluorescence-using-imagej.pdf). For nuclear signals, background was masked and signals from only the nucleus were recorded. For each genotype, $\sim$10 biological samples were quantified. Data are expressed as mean±s.d., and reflect independent experiments and not technical replicates. All statistical analyses were performed using the two-tailed Student's t-test.

## Quantification and statistical analysis

Quantitative analysis of data and graphs were plotted in GraphPad Prism. Each dot in graph represents a data point. Two-tailed unpaired Student's t-test was performed for calculating statistical significance. N numbers and p-values are mentioned in figure legends. Significance of p-values are mentioned as mean±s.d. *P<0.05, **P<0.001 and ***P<0.0001. For instances where there's no statistical significance, "ns" is mentioned. Source data for all the quantitative analyses is included in S1 Data.

## Supporting information

**S1 Fig. Posterior Lobes are divided into compartments by ECM components.** (A-D′.) Trol marks the progenitors and surrounds the niche in the primary lobe (B-B′). Within the posterior lobes, Trol expression creates a large anterior and a small posterior domain (C-D′, magenta asterisk). (E-H′.) Viking reporter expression marks the periphery of the primary lobe (F-F′) and analogous to Trol, divides the secondary and tertiary lobes into two domains (G-H′, magenta asterisk). (I-J′.) The 103908-Gal4 expression is restricted to the posterior domain of both the secondary and tertiary lobe. (K-L′.) 103908-Gal4 does not lineage trace to the anterior half of tertiary lobes (magenta asterisk). In all panels, lymph glands are dissected from third instar larvae (96hr after egg hatching, AEH). Arrows are pointed towards the intercalated pericardial cells within lymph gland lobes. The white arrow marks the pericardial cell between primary and secondary lobes; yellow between secondary and tertiary, and orange marks the last pericardial cell at the end of the LG organ after the tertiary lobe. White dotted lines mark the

lymph gland boundary, while the yellow dotted line marks the aorta, which is between the bi-lobed lymph gland. The genotype of the larvae and labeling on the tissue is described in the panels. Scale bars: 20 μm.
(TIF)

**S2 Fig. Expression analysis of Collier antibody with two of the Col/Knot Gal4 drivers.**
(A-B″.) At 24hr AEH, the entire posterior lobes comprise a single stripe of Col expressing blood cells (marked by antibody) that co-localizes with Kn[13A11]-Gal4 but not pCol-Gal4. Col protein is high in primary lobe PSC but low in medullary zone. (C-F″.) From 48–72 hr AEH, the high Col expression is restricted in the tertiary lobe co-localizing with both Kn[13A11]-Gal4 and pCol-Gal4, while a basal Col antibody labelling is observed in secondary lobes. (G-H″.) At 96hr AEH, high Col expression prevails in the tertiary lobe co-localizing with Kn[13A11]-Gal4 and pCol-Gal4. However, faint Col antibody labelling is detectable in the posterior cells of secondary lobes. (I-J″.) By 120 hr, even when the primary lobe dissociates, the tertiary lobes express high levels of Col that co-localizes with both the drivers. Larval staging is mentioned in the corresponding panels. Arrows are pointed towards the intercalated pericardial cells within lymph gland lobes. The white arrow marks the pericardial cell between primary and secondary lobes; yellow between secondary and tertiary, and orange marks the last pericardial cell at the end of the LG organ after the tertiary lobe. White dotted lines mark the lymph gland boundary, while the yellow dotted line marks the aorta, which is between the bi-lobed lymph gland. The genotype of the larvae and labeling on the tissue is described in the panels. Scale bars: 20 μm.
(TIF)

**S3 Fig. Collier domain harbours both Ultrabithorax (Ubx) positive cells and progenitors and is essential for Ubx expression.** (A-B‴.) At 110hr AEH, the Collier domain in the anterior half of the tertiary lobe harbours both Ubx expressing cells (high Col, Ubx positive) and progenitors (low Col, Ance positive: magenta asterisk). (C-D‴.) At 110hr AEH, Pxn expression can be detected in the Collier negative domain of the tertiary lobe (magenta asterisk). (E-G′.) In Col loss, posterior lobes undergo ectopic differentiation (F-G′) in comparison to control (E-E′). (H-J′.) The loss of Ubx from posterior lobes is observed in Col mutants (I- J′) compared to controls (H-H′). Ubx expression, however, is still unperturbed in cardioblasts (magenta asterisk). In all panels, lymph glands are dissected from third instar larvae (96hr after egg hatching, AEH) unless otherwise mentioned. Arrows are pointed towards the intercalated pericardial cells within lymph gland lobes. The white arrow marks the pericardial cell between primary and secondary lobes; yellow between secondary and tertiary, and orange marks the last pericardial cell at the end of the LG organ after the tertiary lobe. White dotted lines mark the lymph gland boundary, while the yellow dotted line marks the aorta, which is between the bi-lobed lymph gland. The genotype of the larvae and labeling on the tissue is described in the panels. Scale bars: 20 μm.
(TIF)

**S4 Fig. Several bonafide markers of primary lobe PSC express in the anterior half of the tertiary lobe.** (A-A‴) Trol compartmentalizes the Ubx domain. (B-D′) The anterior half of the tertiary lobe expresses Serrate visualized by Ser9.5LacZ (B-B′), LaminC (C-C′), and Fringe Connection (D- D′), all marked by a magenta asterisk. (E) Model summarizing the above findings and of Figs 3 and S4. In all panels, lymph glands are dissected from third instar larvae (96hr after egg hatching, AEH) unless otherwise mentioned. Arrows are pointed towards the intercalated pericardial cells within lymph gland lobes. The white arrow marks the pericardial cell between primary and secondary lobes; yellow between secondary and tertiary, and orange marks the last pericardial cell at the end of the LG organ after the tertiary lobe. White dotted

lines mark the lymph gland boundary, while the yellow dotted line marks the aorta, which is between the bi-lobed lymph gland. The genotype of the larvae and labeling on the tissue is described in the panels. Scale bars: 20 μm.
(TIF)

**S5 Fig. Ubx-Gal4$^{M1}$ expresses exclusively in posterior lobes.** (A-A″) Activation of Lineage tracing construct throughout development by Ubx-Gal4$^{M1}$ marks the entire posterior lobes (A′), live expression is, however, restricted to the anterior half of tertiary lobes (A″) at 96hr AEH. (B-C‴) Ubx-Gal4$^{M1}$ live expression in the tertiary lobe co-localizes with the Ubx antibody at 96hr AEH(B-B′) and zoomed in images (C-C‴). In all panels, lymph glands are dissected from third instar larvae (96hr after egg hatching, AEH). Arrows are pointed towards the intercalated pericardial cells within lymph gland lobes. The white arrow marks the pericardial cell between primary and secondary lobes; yellow between secondary and tertiary, and orange marks the last pericardial cell at the end of the LG organ after the tertiary lobe. White dotted lines mark the lymph gland boundary, while the yellow dotted line marks the aorta, which is between the bi-lobed lymph gland. The genotype of the larvae and labeling on the tissue is described in the panels. Scale bars: 20 μm.
(TIF)

**S6 Fig. Signals from the Ubx zone maintain the posterior lobe progenitors.** (A-D′.) Following the timeline in A, Kn$^{13A11}$-Gal4 was employed to downregulate *sar1* (C-C′) and *hh* (D-D′) from the Ubx zone which resulted in increased differentiation (Pxn) compared to control (B-B′).(E.) Quantification of B-D′, downregulation of *sar1* (n = 8, P-value = 3.51E-04) and *hh* (n = 8, P-value = 1.10849E-06) for primary lobes.(F.) Quantification of B-D′, downregulation of *sar1* (n = 8, P-value = 1.14317E-05) and *hh* (n = 8, P-value = 3.53517E-05) in the secondary lobes. (G.) Quantification of B-D′, downregulation of *sar1* (n = 8, P-value = 0.1779) and *hh* (n = 8, P-value = 0.3506) in tertiary lobe progenitors. (H-K′.) Following the timeline in H, posterior lobe specific driver Ubx-Gal4$^{M1}$ was employed to downregulate *sar1* (J-J′) and *hh* (K-K′) from the Ubx zone resulted in increased differentiation (Pxn) compared to control (I-I′). (L.) Quantification of I-K′, downregulation of *sar1* (Control: n = 15, *sar1-RNAi*: n = 14; P-value = 0.0719) and *hh* (Control: n = 15, *hh-RNAi*: n = 15; P-value = 0.0970) for primary lobes. (M.) Quantification of I-K′, downregulation of *sar1* (Control: n = 15, *sar1-RNAi*: n = 13; P-value = 1.20E-03) and *hh* (Control: n = 15, *hh-RNAi*: n = 15; P-value = 7.70E-04) in the secondary lobes. (N.) Quantification of I-K′, downregulation of *sar1* (Control: n = 15, *sar1-RNAi*: n = 11; P-value = 0.0190) and *hh* (Control: n = 15, *hh-RNAi*: n = 15; P-value = 4.66E-03) in tertiary lobe progenitors. In all panels, lymph glands are dissected from third instar larvae (96hr after egg hatching, AEH). Arrows are pointed towards the intercalated pericardial cells within lymph gland lobes. The white arrow marks the pericardial cell between primary and secondary lobes; yellow between secondary and tertiary, and orange marks the last pericardial cell at the end of the LG organ after the tertiary lobe. White dotted lines mark the lymph gland boundary, while the yellow dotted line marks the aorta, which is between the bi-lobed lymph gland. The genotype of the larvae and labeling on the tissue is described in the panels. Scale bars: 20 μm. Two-tailed unpaired Student's t-test was performed for calculating statistical significance. Data are mean±s.d. *P<0.05, **P<0.001 and ***P<0.0001.
(TIF)

**S7 Fig. Posterior lobe specific Ubx upregulation prevents posterior lobe progenitors from responding to immune challenge.** (A-C′.) Ectopic differentiation (Pxn) is observed upon wasp infection in primary and secondary lobes (B-B′) compared to uninfected control (A-A′). Overexpression of Ubx from the tertiary lobe utilizing posterior lobe specific driver Ubx-

Gal4$^{M1}$ suppresses the immune response from the posterior lobes (C-C′, magenta asterisk). (D-F′.) Lamellocyte induction (β-PS) is observed upon wasp infection in primary and secondary lobes (E-E′) compared to uninfected control (D-D′). Overexpression of Ubx from the tertiary lobe utilizing posterior lobe specific driver Ubx-Gal4$^{M1}$ reduces the number of lamellocytes formed in the posterior lobes (F-F′). (G.) Quantification of the differentiation index of secondary lobe (A-C′.) {Uninfected vs Infected: n = 15, P-value = 4.676E-07}, {Uninfected vs Ubx overexpression: n = 15, P-value = 6.341E-05} and {Infected vs Ubx overexpression: n = 15, P-value = 2.342E-03}. (H.) Quantification of the differentiation index of tertiary lobe (A-C′.) {Uninfected vs Infected: n = 15, P-value = 5.558E-03}, {Uninfected vs Ubx overexpression: n = 15, P-value = 0.146} and {Infected vs Ubx overexpression: n = 15, P-value = 0.0455}. (I.) Quantification of total hemocyte pool in secondary lobe (A-F′) {Uninfected vs Infected: n = 15, P-value = 2.20E-07}, {Uninfected vs Ubx overexpression: n = 15, P-value = 4.19E-03} and {Infected vs Ubx overexpression: n = 15, P-value = 1.855E-03}. (J.) Quantification of the progenitor pool in tertiary lobe (A-F′.) {Uninfected vs Infected: n = 15, P-value = 1.283E-03}, {Uninfected vs Ubx overexpression: n = 15, P-value = 0.0652} and {Infected vs Ubx overexpression: n = 15, P-value = 0.0164}. (K.) Quantification of lamellocytes in primary lobe (D-F′) {Uninfected vs Infected: n = 15, P-value = 1.893E-08}, {Uninfected vs Ubx overexpression n = 15, P-value = 8.891E-05} and {Infected vs Ubx overexpression: n = 15, P-value = 0.0148}. (L.) Quantification of lamellocytes in secondary lobe (D-F′) {Uninfected vs Infected: n = 15, P-value = 1.570E-06}, {Uninfected vs Ubx overexpression n = 15, P-value = 0.0103} and {Infected vs Ubx overexpression: n = 15, P-value = 2.781E-04}. (M.) Quantification of lamellocytes in tertiary lobe(D-F′.) {Uninfected vs Infected: n = 15, P-value = 3.091E-03}, {Uninfected vs Ubx overexpression n = 15, P-value = 0.1643} and {Infected vs Ubx overexpression: n = 15, P-value = 0.0182}. In all panels, lymph glands are dissected from third instar larvae (96hr after egg hatching, AEH). Arrows are pointed towards the intercalated pericardial cells within lymph gland lobes. The white arrow marks the pericardial cell between primary and secondary lobes; yellow between secondary and tertiary, and orange marks the last pericardial cell at the end of the LG organ after the tertiary lobe. White dotted lines mark the lymph gland boundary, while the yellow dotted line marks the aorta, which is between the bi-lobed lymph gland. The genotype of the larvae and labeling on the tissue is described in the panels. Scale bars: 20 μm. Two-tailed unpaired Student's t-test was performed for calculating statistical significance. Data are mean±s.d. *P<0.05, **P<0.001 and ***P<0.0001.
(TIF)

**S8 Fig. Ubx levels drop upon wasp parasitism.** (A-B′.) Ubx levels drop substantially upon wasp infection induced immune challenge (B-B′) in comparison to un-infected control (A-A′). Note that there is no change in Ubx levels of neighbouring pericardial cells and cardioblasts. (C.) Statistical analysis reveals the significance of drop in Ubx levels (n = 22, P-value = 5.011E-09). In all panels, lymph glands are dissected from third instar larvae (96hr after egg hatching, AEH). Arrows are pointed towards the intercalated pericardial cells within lymph gland lobes. The white arrow marks the pericardial cell between primary and secondary lobes; yellow between secondary and tertiary, and orange marks the last pericardial cell at the end of the LG organ after the tertiary lobe. White dotted lines mark the lymph gland boundary, while the yellow dotted line marks the aorta, which is between the bi-lobed lymph gland. The genotype of the larvae and labeling on the tissue is described in the panels. Scale bars: 20 μm. Two-tailed unpaired Student's t-test was performed for calculating statistical significance. Data are mean ±s.d. *P<0.05, **P<0.001 and ***P<0.0001.
(TIF)

**S9 Fig. Ubx over-expression leads to an increase in Hedgehog levels.** (A-C″) Upregulation of Ubx via posterior lobe specific driver Ubx-Gal4[M1] (C-C″), following the scheme in A, leads to a substantial increase in *hedgehog* levels in comparison to control (B-B″). (D.) Quantification of (B-C″) in terms of *hhF4fGFP* signal (n = 14, P-value = 2.868E-07). (E-G″) Upregulation of Ubx via Kn[13A11]-Gal4 (G-G″), following the scheme in E, leads to a significant increase in *hedgehog* levels in comparison to control (F-F″). (H.) Quantification of (F-G″) in terms of *hhF4fGFP* signal (n = 10, P-value = 3.277E-06). In all panels, lymph glands are dissected from third instar larvae (96hr after egg hatching, AEH). Arrows are pointed towards the intercalated pericardial cells within lymph gland lobes. The white arrow marks the pericardial cell between primary and secondary lobes; yellow between secondary and tertiary, and orange marks the last pericardial cell at the end of the LG organ after the tertiary lobe. White dotted lines mark the lymph gland boundary, while the yellow dotted line marks the aorta, which is between the bi-lobed lymph gland,. The genotype of the larvae and labeling on the tissue is described in the panels. Scale bars: 20 μm. Two-tailed unpaired Student's t-test was performed for calculating statistical significance. Data are mean±s.d. *P<0.05, **P<0.001 and ***P<0.0001.
(TIF)

**S10 Fig. Affecting the functionality of the Ubx domain elicits extensive differentiation post wasp infection.** (A-H′.) Differentiation in response to wasp parasitism (B-B′) increases further when the functionality of the Ubx domain is perturbed via downregulation of *hh* (D-D′), *sar1* (F-F′) and *Ubx* (H-H′) in comparison to their respective controls (A-A′), (C-C′), (E-E′) and (G-G′). (I.) Quantification of (A-H′) reveals the statistical significance of increase in differentiation of secondary lobes upon wasp infection. (Control: Un-infected vs Infected, n = 10, P-value = 6.886E-08; *hh-RNAi*: Un-infected vs Infected, n = 10, P-value = 3.947E-05; *sar1-RNAi*: Un-infected vs Infected, n = 10, P-value = 0.0144; *Ubx-RNAi*: Un-infected vs Infected, n = 10, P-value = 8.42E-03). (J.) Quantification of (A-H′) reveals the statistical significance of increase in differentiation of tertiary lobes upon wasp infection. (Control: Un-infected vs Infected, n = 10, P-value = 0.033; *hh-RNAi*: Un-infected vs Infected, n = 10, P-value = 0.237; *sar1-RNAi*: Un-infected vs Infected, n = 10, P-value = 0.088; *Ubx-RNAi*: Un-infected vs Infected, n = 10, P-value = 0.919). In all panels, lymph glands are dissected from third instar larvae (96hr after egg hatching, AEH). Arrows are pointed towards the intercalated pericardial cells within lymph gland lobes. The white arrow marks the pericardial cell between primary and secondary lobes; yellow between secondary and tertiary, and orange marks the last pericardial cell at the end of the LG organ after the tertiary lobe. White dotted lines mark the lymph gland boundary, while the yellow dotted line marks the aorta, which is between the bi-lobed lymph gland. The genotype of the larvae and labeling on the tissue is described in the panels. Scale bars: 20 μm. Two-tailed unpaired Student's t-test was performed for calculating statistical significance. Data are mean±s.d. *P<0.05, **P<0.001 and ***P<0.0001.
(TIF)

**S11 Fig. Ubx zone upregulates Dif and ROS upon wasp infection.** (A-D′.) D4-LacZ (Dif) significantly increases in Antp zone as well as Ubx zone (B-B′ (mild response), C-C′ (strong response)) in comparison to control upon wasp infection (A-A′). Upregulation of Ubx via pCol-Gal4 suppresses Dif levels during wasp infection (D-D′), indicated by magenta asterisk. (E.) Quantitative analysis of D4-LacZ levels in Antp zone (Un-infected vs Infected: n = 13, P-value = 6.86143E-10). The increase in intensity is unaffected even upon Ubx overexpression via pCol-Gal4 (Un-infected vs Ubx overexpression: n = 13, P-value = 1.431E-09) and (Infected vs Ubx overexpression: n = 13, P-value = 0.320). (F.) Wasp infection induces D4 (Dif) in Ubx domain (Un-infected vs Infected: n = 13, P-value = 2.54857E-08) and its levels are rescued upon Ubx overexpression (Un-infected vs Ubx overexpression n = 13, P-value = 6.643E-05)

and (Infected vs Ubx overexpression: n = 13, P-value = 4.813E-08). (G-H″.) Wasp infection induces gstD-GFP in Ubx zone (H-H″), compared to control (G-G″), marked by a magenta asterisk. (I.) Quantitation of G-H″ (Antp domain: n = 10, P-value = 3.08E-04; Ubx domain: n = 10, P-value = 3.67E-04). (J-K′.) Wasp infection generates ROS (DHE, K-K′) in the Ubx domain compared to uninfected (J-J′), marked by a magenta asterisk. (L.) Quantitative analysis of J-K′ (Antp domain: n = 8 (Control), n = 10 (infected), P-value = 1.58E-03; Ubx domain: n = 8 (Control), n = 10 (infected), P-value = 6.77E-04). In all panels, lymph glands are dissected from third instar larvae (96hr after egg hatching, AEH). Arrows are pointed towards the intercalated pericardial cells within lymph gland lobes. The white arrow marks the pericardial cell between primary and secondary lobes; yellow between secondary and tertiary, and orange marks the last pericardial cell at the end of the LG organ after the tertiary lobe. White dotted lines mark the lymph gland boundary, while the yellow dotted line marks the aorta, which is between the bi-lobed lymph gland. The genotype of the larvae and labeling on the tissue is described in the panels. Scale bars: 20 μm. Two-tailed unpaired Student's t-test was performed for calculating statistical significance. Data are mean±s.d. *P<0.05, **P<0.001 and ***P<0.0001.
(TIF)

**S1 Table. Fly Stocks used for the current study.**
(DOCX)

**S1 Data. This excel file contains the quantitative data used for panels presented in Figs 3–7 and supplementary S6–S11 Figs.**
(XLSX)

## Acknowledgments

We thank L. Cooley, M.A. Yoo, U. Banerjee, R.A. Schulz, S Thor, M. Gonzales-Gaitan, A. Sharma, R.A. Schulz, M. Crozatier, D. Bohmann, E. Sanchez-Herrero, J. Benito-Sipos, L.S. Shashidhara and T. Mukherjee for reagents. We thank Saikat Ghosh and Adyasha Nayak for their help with the initial part of the study. We thank all members of the two laboratories for their valuable input. We thank IISER Mohali's Confocal Facility, the Bloomington *Drosophila* Stock Center at Indiana University, the Vienna *Drosophila* Resource Center, Kyoto Stock Center and the Developmental Studies Hybridoma Bank at the University of Iowa for flies and antibodies. Models were made using BioRender. Adobe Photoshop Elements 2019 was used to prepare Panels. Fiji (ImageJ) was used for image processing and graphs were prepared using GraphPad.

## Author Contributions

**Conceptualization:** Lolitika Mandal.

**Data curation:** Aditya Kanwal.

**Formal analysis:** Aditya Kanwal, Pranav Vijay Joshi, Sudip Mandal, Lolitika Mandal.

**Funding acquisition:** Lolitika Mandal.

**Investigation:** Aditya Kanwal, Lolitika Mandal.

**Methodology:** Aditya Kanwal, Lolitika Mandal.

**Project administration:** Lolitika Mandal.

**Supervision:** Lolitika Mandal.

**Validation:** Aditya Kanwal, Pranav Vijay Joshi.

**Visualization:** Aditya Kanwal, Pranav Vijay Joshi, Lolitika Mandal.

**Writing – original draft:** Aditya Kanwal, Lolitika Mandal.

**Writing – review & editing:** Aditya Kanwal, Sudip Mandal, Lolitika Mandal.

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
