## [Decision Letter · Decision Letter 0]

8 Feb 2021

Dear Dr Mandal,

Thank you very much for submitting your Research Article entitled 'Ubx-Collier signaling cascade maintains blood progenitors in the posterior lobes of the Drosophila larval lymph gland' to PLOS Genetics.

The manuscript was fully evaluated at the editorial level and by independent peer reviewers. The reviewers appreciated the attention to an important problem, but raised some substantial concerns about the current manuscript. Based on the reviews, we will not be able to accept this version of the manuscript, but we would be willing to review a much-revised version. We cannot, of course, promise publication at that time.

If you decide to revise the manuscript for further consideration at PLOS Genetics, please aim to resubmit within the next 60 days, unless it will take extra time to address the concerns of the reviewers, in which case we would appreciate an expected resubmission date by email to plosgenetics@plos.org.

[LINK]

We are sorry that we cannot be more positive about your manuscript at this stage. Please do not hesitate to contact us if you have any concerns or questions.

Yours sincerely,

Hongyan Wang, Ph.D.

Associate Editor

PLOS Genetics

Gregory P. Copenhaver

Editor-in-Chief

PLOS Genetics

Reviewer's Responses to Questions

**Comments to the Authors:**

Reviewer #1: This manuscript characterized the cell heterogeneity in the posterior lobes of Drosophila lymph lode and found a Ubx+Ance- zone might function as a potential niche regulate the posterior lobe progenitors. While this general finding is interesting, I find that the characterization of the niche function somewhat weak and casual. Figure 2 and 6 seem to diverge from the major point and logic flow and I think they should be left as supplemental figures.

Specific points are listed as follows:

1. Figure 1A: Please mark the anterior/primary lobe and posterior/secondary/tertiary lobes clearly in the schematic view.

2. Figure 1B: why is Trol staining in posterior lobes significantly different from the anterior lobe (absent from the cells in the center)? The staining looks that Trol is on the periphery of these lobes so it seems to say it is uniformly stained among the cells.

3. Figure 1C-F, why is the morphology of secondary and tertiary lobes differ a lot from 1B? The secondary lobes in these panels contain only one lobe while in Figure 1B it contains 2 lobes.

4. Figure 1H: Upon wasp infection, is there a shrinkage of the Ance- domain of the tertiary lobe? Moreover, I don’t see why the data shown rule out the possibility Pxn+ in the figure didn’t rise from Ance- cells.

5. Figure 1I: The schematic figure here seems completely unnecessary.

6. Figure S1D: is it two domains? The figure looks like three domains.

7. Figure 1 title: Please rephrase “Posterior Lobes are a heterogeneous bunch of cells” to academic language.

8. In Figure 2, please explain how the arrows are drawn to define secondary vs tertiary lobe.

9. In Figure S2, I find the choice of staging to 110h AEH incompatible with the staging in Figure 1, if the authors want to show that Ance- region previously describe in Figure 1 is Ubx+, use the same staging is essential.

10. Figure 4A-J, Ubx and Pxn in the same channel is very difficult for readers to visualize in the figure. Define how differentiation index is calculated.

11. I don’t understand what is sar1 and why downregulating it blocks niche signaling. Moreover, the result in Figure S4 seems important enough to be incorporated to Figure 4.

12. What is Figure 4K and L? There is no explanation in the article.

13. Figure 5, what is the timing of this experiment? I find overexpression of Ubx in a domain where its expression is already high somewhat puzzling. What is the phenotype upon infection when Ubx or hh is knocked down?

Minor point:

Line 97 “unique zonation signifying distinct cell types within these lobes” please rephrase for clarity.

Reviewer #2: Review uploaded as an attachment

Reviewer #3: The lymph gland of Drosophila plays a significant regulatory role in larval hematopoiesis. Several functional cellular compartments and regulatory factors have been identified, however the studies focused mainly on the anterior lobes of the organ. Developmental studies suggested that the posterior lobes and their regulation is different from those of the primary lobes, but their regulation, their role in hematopoiesis and in the immune response has not been explored. The Authors of the presented studies reveal the functional heterogeneity of hematopoietic progenitors residing in these lobes and identify regulatory factors acting in the separate niches and responsible for the regulation of the distinct functional progenitor populations.

Minor comments:

Lines 65 and 691: SHRESTHA and GATEFF are in capital

Cited articles missing from “References” section:

line 465 Kurucz et al., 2007

line 467 Shirras , Hurst et al., 2003

**Have all data underlying the figures and results presented in the manuscript been provided?**

Reviewer #1: Yes

Reviewer #2: Yes

Reviewer #3: Yes

PLOS authors have the option to publish the peer review history of their article (what does this mean?). If published, this will include your full peer review and any attached files.

Reviewer #1: No

Reviewer #2: **Yes: **Alain VINCENT

Reviewer #3: No

---

## [Decision Letter · Decision Letter 1]

14 Jun 2021

Dear Dr Mandal,

Thank you very much for submitting your Research Article entitled 'Ubx-Collier signaling cascade maintains blood progenitors in the posterior lobes of the Drosophila larval lymph gland' to PLOS Genetics.

The manuscript was fully evaluated at the editorial level and by independent peer reviewers. The reviewers appreciated the attention to an important topic but identified some concerns that we ask you address in a revised manuscript

We therefore ask you to modify the manuscript according to the review recommendations. Your revisions should address the specific points made by each reviewer.

[LINK]

Yours sincerely,

Hongyan Wang, Ph.D.

Associate Editor

PLOS Genetics

Gregory P. Copenhaver

Editor-in-Chief

PLOS Genetics

Reviewer's Responses to Questions

**Comments to the Authors:**

Reviewer #1: While the manuscript has been improved, I am not yet convinced that the newly-identified Ubx zone functions as a signaling center. The functional assays of hh RNAi and sar1 RNAi using Ubx-Gal4 in this zone, provided in supplemental figure 6 and 10 seem to show rather minor and puzzling effects. In supplemental figure 6, loss of hh lead to increase of differentiation in secondary and tertiary lobes according to the authors. Why would loss of hh upon wasp infection lead to decrease of differentiation in the secondary and teritary zone. What is the phenotype of Hh over expression in the Ubx zone?

Reviewer #2: Manuscript Number PGENETICS-D-20-01903R1

Comments for the authors

The revised manuscript by Kanwal et al., has satisfactorily answered most of my suggestions and criticisms. I than the authors for this. I feel that the revisions made have significantly improved their ms.

Considering some main points I previously raised, revisions include : (1) A comparison between the two col/kn drivers which revealed a different temporality of expression (Fig.S2) ; (3) most importantly, an extension of the lineage and RNAi KD experiments using a driver expressed in secondary lobes but not the PSC (Fig. 4 and Fig.S5) ; these experiments nicely confirm the previous main conclusions of the authors, while revealing that the PSC contributes to the maintenance of progenitors in posterior lobes, a conclusion not reached in the initial version.

Before publication, a few more corrrections are needed.

Major.

Point 2, my concern about the KnGal4 nomenclature was not adressed. I would prefer not to compromise on this point. As I already wrote, the authors improperly use Kn-Gal4 to design a specific KnGal4 GMR line, line 13A11, characterised under this name to be expressed in the LG (Tokusimi et al., 2017). Many Kn/col enhancer and driver lines have previously been characterised, starting by wing enhancers (incidentally under Ubx control; Hersh and Carroll, Development 2005 PMID 15753212). I strongly suggest to refer to the specific KnGal4 line used by autors by its original name, GMR13A11 or KnGal413A11 (13A11 superscript). This may seem a detail but connecting the present data to previous data obtained in other labs and exploited here, would be fair and not confusing.

Minor

- Contribution, both of the PSC and PSLC to maintenance of secondary lobe progenitors should be better emphasized in the discussion section, with a more moderate conclusion, line 465.

- Fig.S1 A-C. there is a discrepancy/confusion between the text of legend S1 A-H’ : Trol marks the progenitors in the primary lobe ; Viking expression marks the periphery of the primary lobe and the main text lines lines 257-264 . Similar enrichment of Trol in the niche of the primary lobes (25). Viking surrounding of the primary niche was first noted by Krzemien et al, 2010 (11) and Trol surrounding (and not enrichement) by Grigorian et al., 2013 (25). Please try to homogenise.

- Fig 2B-B’’ text line 199 Kn should be KnGal4(13A11)

- Fig.4 J-J’ Not only Ubx, but also Ubx(M1)-Gal4 seems down-regulated ; Does this reflect Ubx autoregulation of Ubx(M1) ? Please comment, if necessary.

Text : Check whether the phrasing is accurate for :

lines 214-219 ; 210-212, Ux ?,Col ? regression or regression of both ?; 334-336 (Since ?) and 338 ; 360-363 ; 421-424 ; 467-468.

Lines 445-449 should be moved to discussion.

Reviewer #3: This is an improved version of the manuscript.

**Have all data underlying the figures and results presented in the manuscript been provided?**

Reviewer #1: Yes

Reviewer #2: Yes

Reviewer #3: Yes

PLOS authors have the option to publish the peer review history of their article (what does this mean?). If published, this will include your full peer review and any attached files.

Reviewer #1: No

Reviewer #2: **Yes: **Alain Vincent

Reviewer #3: No

---

## [Decision Letter · Decision Letter 2]

12 Jul 2021

Dear Dr Mandal,

We are pleased to inform you that your manuscript entitled "Ubx-Collier signaling cascade maintains blood progenitors in the posterior lobes of the Drosophila larval lymph gland" has been editorially accepted for publication in PLOS Genetics. Congratulations!

Yours sincerely,

Hongyan Wang, Ph.D.

Associate Editor

PLOS Genetics

Gregory P. Copenhaver

Editor-in-Chief

PLOS Genetics

Comments from the reviewers (if applicable):

Reviewer's Responses to Questions

**Comments to the Authors:**

Reviewer #1: The authors have addressed my questions and I support its publication.

Reviewer #2: Thanks to the authors for their answers to my comments and suggestions.

No further comments

**Have all data underlying the figures and results presented in the manuscript been provided?**

Reviewer #1: Yes

Reviewer #2: Yes

PLOS authors have the option to publish the peer review history of their article (what does this mean?). If published, this will include your full peer review and any attached files.

Reviewer #1: No

Reviewer #2: **Yes: **Alain VINCENT

**Data Deposition**

http://datadryad.org/submit?journalID=pgenetics&manu=PGENETICS-D-20-01903R2

**Press Queries**

---

## [Editor Report · Acceptance letter]

4 Aug 2021

PGENETICS-D-20-01903R2 

Ubx-Collier signaling cascade maintains blood progenitors in the posterior lobes of the Drosophila larval lymph gland 

Dear Dr Mandal, 

We are pleased to inform you that your manuscript entitled "Ubx-Collier signaling cascade maintains blood progenitors in the posterior lobes of the Drosophila larval lymph gland" has been formally accepted for publication in PLOS Genetics! Your manuscript is now with our production department and you will be notified of the publication date in due course.

With kind regards,

Melanie Wincott

PLOS Genetics

On behalf of:
